# Evolution of multifunctionality through a pleiotropic substitution in the innate immune protein S100A9

Joseph L Harman[1,2], Andrea N Loes[1,2], Gus D Warren[1,2], Maureen C Heaphy[1,2], Kirsten J Lampi[3], Michael J Harms[1,2]*

[1]Department of Chemistry and Biochemistry, University of Oregon, Eugene, United States; [2]Institute of Molecular Biology, University of Oregon, Eugene, United States; [3]Oregon Health & Science University, Portland, United States

**Abstract** Multifunctional proteins are evolutionary puzzles: how do proteins evolve to satisfy multiple functional constraints? S100A9 is one such multifunctional protein. It potently amplifies inflammation via Toll-like receptor four and is antimicrobial as part of a heterocomplex with S100A8. These two functions are seemingly regulated by proteolysis: S100A9 is readily degraded, while S100A8/S100A9 is resistant. We take an evolutionary biochemical approach to show that S100A9 evolved both functions and lost proteolytic resistance from a weakly proinflammatory, proteolytically resistant amniote ancestor. We identify a historical substitution that has pleiotropic effects on S100A9 proinflammatory activity and proteolytic resistance but has little effect on S100A8/S100A9 antimicrobial activity. We thus propose that mammals evolved S100A8/S100A9 antimicrobial and S100A9 proinflammatory activities concomitantly with a proteolytic 'timer' to selectively regulate S100A9. This highlights how the same mutation can have pleiotropic effects on one functional state of a protein but not another, thus facilitating the evolution of multifunctionality.

*For correspondence: harms@uoregon.edu

Competing interests: The authors declare that no competing interests exist.

## Introduction

The innate immune system uses a small number of multifunctional proteins to respond to diverse immune challenges (*Lee et al., 2018*; *Auvynet and Rosenstein, 2009*; *Postel et al., 2010*; *Gudmundsson and Agerberth, 1999*). Multifunctional immune proteins are critical for pathogen defense, (*Auvynet and Rosenstein, 2009*; *Postel et al., 2010*; *Gudmundsson and Agerberth, 1999*) shaping host-associated microbial communities, (*Singh et al., 2016*) and well-regulated tissue growth (*Säemann et al., 2009*; *Bals and Wilson, 2003*; *Leask and Abraham, 2003*). They also drive pathological inflammation in disease, including autoimmune disorders, cancer, and cardiovascular disease (*Ehrchen et al., 2009*; *Salama et al., 2008*; *Shabani et al., 2018*; *Gruden et al., 2016*; *Peng et al., 2018*). These multifunctional proteins raise both mechanistic and evolutionary questions. How can one protein sequence satisfy the multiple constraints imposed by having multiple functions? How can multiple functions evolve in one protein when, as a result of multifunctionality, each mutation likely has pleiotropic effects? (*Lee et al., 2018*; *Bomblies and Doebley, 2006*; *Stearns, 2010*; *Paaby and Rockman, 2013*).

One such multifunctional protein is S100A9 (A9), a small, soluble protein found at high concentrations in the extracellular space during an inflammatory response (*Berntzen and Fagerhol, 1990*). It has at least two key immune functions. As a homodimer, A9 potently activates inflammation via Toll-like receptor 4 (TLR4) (*Vogl et al., 2012*; *Källberg et al., 2012*; *Duan et al., 2018*; *Shepherd et al., 2006*; *Schiopu and Cotoi, 2013*; *Laouedj et al., 2017*; *He et al., 2016*; *Gao et al., 2015*; *Tsai et al., 2014*; *Lee et al., 2016*; *Björk et al., 2009*; *Kang et al., 2015*; *Stríz and Trebichavský,*

**eLife digest** A single protein sometimes does multiple jobs. For instance, our immune system uses a small number of multipurpose proteins to respond quickly to a large number of threats. One example is the protein S100A9. It acts as an antimicrobial by preventing microbes from getting the nutrients they need, while also stimulating inflammation by inducing the release of molecules that recruit white blood cells.

S100A9, like all proteins, is made up of a chain of small building blocks. These building blocks interact with each other and with other molecules in the environment. The sequence of the building blocks thus determines what jobs the protein can do. Therefore, a single change to the sequence of building blocks can have a dramatic effect: one change might render the protein faulty, while another change might allow it to do a new job.

Proteins face similar challenges humans do when trying to do several things at once. A person driving a car while using their phone will not do either task well. Likewise, a protein that does two jobs faces challenges a single-purpose protein does not.

Harman et al. were interested in how S100A9 was able to evolve and maintain its dual functionality, despite this potential problem. They started by asking when S100A9 acquired its two purposes. They measured the antimicrobial and inflammatory activity of S100A9 proteins from humans, mice and opossums. The activities of S100A9 in these species was similar, suggesting that S100A9 acquired its different jobs in the ancestor of mammals, some 160 million years ago.

Next, Harman et al. computationally reconstructed ancestral forms of S100A9 by comparing hundreds of similar proteins and building an evolutionary tree. They then measured the antimicrobial and inflammatory activity of these ancestral proteins. By comparing the last ancestor that did not have these activities to the first ancestor that did, they identified the sequence changes that gave S100A9 its dual activity. Importantly, these changes are located in separate regions of the protein, meaning they could occur independently, without affecting each other. Further, the same sequence change that converted S100A9 into an inflammatory signal also introduced a mechanism to regulate this activity.

The results suggest that a small number of sequence changes – or even a single change – can make a protein more versatile. This means that evolving multipurpose proteins may not be as difficult as is often thought.

*2004*). As a heterocomplex with S100A8 (A8/A9, also known as calprotectin), it is antimicrobial (*Figure 1a*; *Besold et al., 2018a*; *Hadley et al., 2018*; *Damo et al., 2013*; *Clark et al., 2016*; *Besold et al., 2018b*; *Nakashige et al., 2016*; *Hayden et al., 2013*; *Nakashige et al., 2015*; *Liu et al., 2012*; *Brunjes Brophy et al., 2013*; *Brophy et al., 2012*; *Baker et al., 2017*; *Gagnon et al., 2015*; *Nisapakultorn et al., 2001*). A9 exacerbates endotoxin-induced shock in mice (*Vogl et al., 2007*). Both A9 and A8/A9 are primary biomarkers for many human inflammatory diseases (*Vogl et al., 2014*; *Hara et al., 2012*; *Obry et al., 2014*; *Horvath et al., 2016*; *Huang et al., 2015*). Further, dysregulation of A9 is associated with various cancers, pulmonary disorders, and Alzheimer's disease (*Källberg et al., 2012*; *Vogl et al., 2014*; *Hara et al., 2012*; *Obry et al., 2014*; *Horvath et al., 2016*; *Huang et al., 2015*; *Kim et al., 2009*; *Averill et al., 2012*). Understanding the mechanisms by which A9 performs its innate immune functions is critical for developing treatments for A9-mediated diseases.

The mechanism of A8/A9 antimicrobial activity is well established: it sequesters a variety of transition metals through both a hexahistidine site and a His$_3$Asp site formed at the A8/A9 heterodimer interface, thereby limiting the concentrations of essential microbial nutrients in the extracellular space (*Besold et al., 2018a*; *Hadley et al., 2018*; *Damo et al., 2013*; *Clark et al., 2016*; *Besold et al., 2018b*; *Nakashige et al., 2016*; *Hayden et al., 2013*; *Nakashige et al., 2015*; *Liu et al., 2012*; *Brunjes Brophy et al., 2013*; *Brophy et al., 2012*; *Baker et al., 2017*; *Gagnon et al., 2015*; *Nisapakultorn et al., 2001*). Other S100 proteins exert weaker antimicrobial activity via the His$_3$Asp site, which has lower metal-binding affinity and binds fewer types of transition metals than the A8/A9 hexahistidine site (*Nakashige et al., 2016*; *Hayden et al., 2013*; *Nakashige et al., 2015*; *Liu et al., 2012*; *Brunjes Brophy et al., 2013*; *Bozzi and Nolan, 2020*;

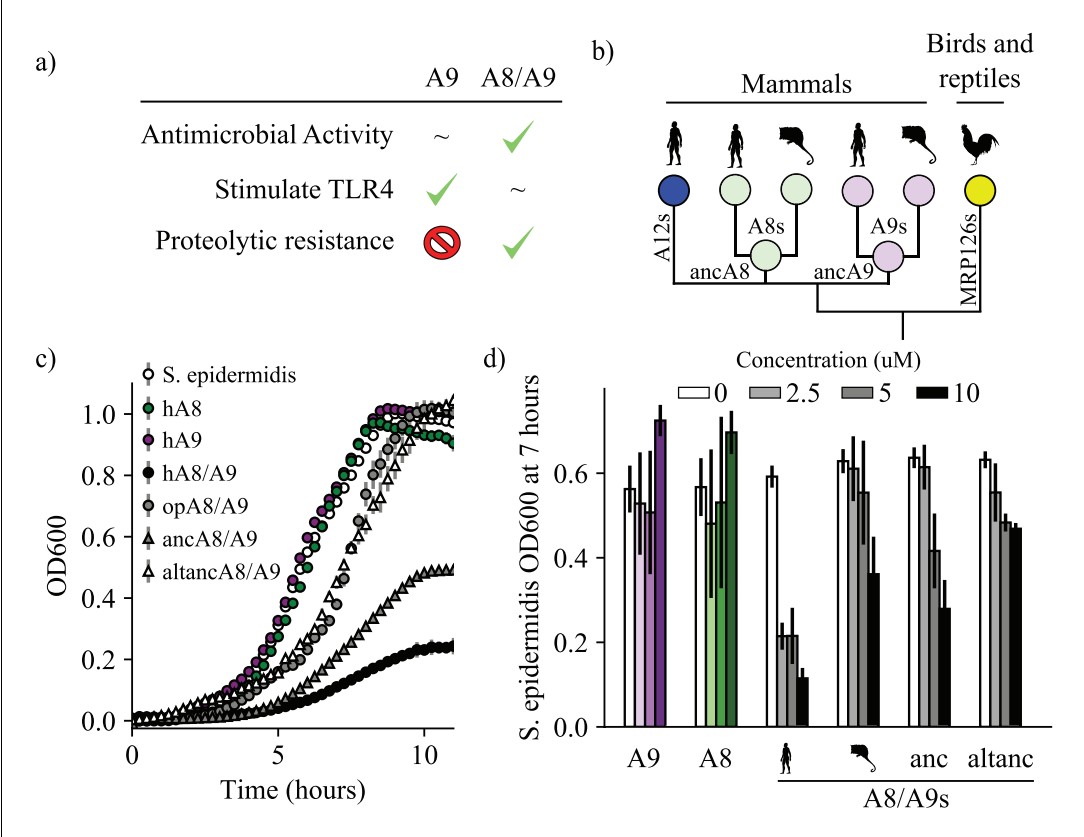

**Figure 1.** A9s evolved to form the antimicrobial A8/A9 complex early in mammals. (**a**) Table of A9 and A8/A9 properties. '~" represents weak or ambiguously characterized function, check marks and red 'X' represent confirmed property (check) or lack thereof ('x'). (**b**) Schematic of previously published S100 protein tree. Colored nodes represent single protein sequences. Species cartoons shown are human, opossum, and chicken. Ancestrally reconstructed protein nodes are labeled. Branch lengths not to scale. (**c**) Representative growth curves for *Staphylococcus epidermidis* in the presence or absence of 10 µM S100 proteins. Each point represents optical density at 600 nm. *S. epidermidis* growth alone and in the presence of modern proteins are shown as circles, growth in the presence of ancestrally reconstructed proteins shown as triangles. Error bars are standard deviation of three technical replicates. (**d**) Percent of untreated *S. epidermidis* growth at 12 hr with S100 protein treatments. Data are average of three biological replicates. Error bars are standard error of the mean. Species cartoon labels are the same as in (**b**).

The online version of this article includes the following source data and figure supplement(s) for figure 1:

**Source data 1.** Antimicrobial activity of S100 proteins.
**Figure supplement 1.** The residues composing the A8/A9 hexastidine site are conserved across mammalian and ancestrally reconstructed A8s and A9s.
**Figure supplement 2.** Analysis of protein oligomeric state using SECMALS.
**Figure supplement 3.** Bacterial growth in the presence of S100 proteins.
**Figure supplement 4.** Statistics for ancestrally reconstructed proteins.
**Figure supplement 5.** Secondary structure characterization of ancestrally reconstructed S100s by CD spectroscopy.

*Cunden and Nolan, 2018*). In contrast, the proinflammatory mechanism of A9 is not well understood. A9 acts as a Damage-Associated Molecular Pattern (DAMP), activating NF-κB and other cytokines through Toll-like receptor 4 (TLR4) (*Vogl et al., 2012*; *Källberg et al., 2012*; *Duan et al., 2018*; *Shepherd et al., 2006*; *Schiopu and Cotoi, 2013*; *Laouedj et al., 2017*; *He et al., 2016*; *Gao et al., 2015*; *Tsai et al., 2014*; *Lee et al., 2016*; *Björk et al., 2009*; *Kang et al., 2015*; *Chen et al., 2015*). The interaction interface(s), affinity, and stoichiometry for the A9/TLR4 interaction are not known. A small region of A9 has been suggested to form part of the A9/TLR4 binding surface, (*Vogl et al., 2018*) but no mutant of A9 has been identified that substantially compromises its activation of TLR4.

An additional layer of A9 immune function is that A9 and A8/A9 are thought to be regulated in the extracellular milieu by proteases. Neutrophils release multiple proteases along with A9 at sites

of inflammation that regulate the inflammatory response (*Henry et al., 2016*; *Stapels et al., 2015*; *Kessenbrock et al., 2011*; *Heutinck et al., 2010*; *Janoff, 1972*; *Jerke et al., 2015*). A9 is very susceptible to proteolytic degradation, while A8/A9 is highly resistant (*Figure 1a*; *Nacken and Kerkhoff, 2007*; *Riva et al., 2013*). Proteolysis may serve to purge proinflammatory A9 from sites of inflammation and thus selectively enrich for antimicrobial A8/A9. There may even be a direct, functional link between A9 proteolytic degradation and inflammation. Proteolytic fragments of A9 are sufficient to activate TLR4, (*Vogl et al., 2018*) and proinflammatory stimuli are thought to stabilize A9 homodimers against proteolytic degradation (*Riva et al., 2013*). Directly testing the relationship between A9 proteolytic susceptibility and proinflammatory activity, however, has been challenging. There is no obvious way to selectively increase the proteolytic resistance of A9 and test its effect on A9 activation of TLR4, making it difficult to understand the relationship, if any, between these two functions.

We took an evolutionary biochemical approach to mechanistically dissect the evolution of A9 innate immune functions. Using phylogenetics, ancestral sequence reconstruction (ASR), and biochemical studies, we show that A9s evolved to form proteolytically resistant, antimicrobial A8/A9 complexes in early mammals. We find that A9 homodimers gained proinflammatory activity and lost proteolytic resistance in the ancestor of therian mammals from a weakly proinflammatory, proteolytically resistant amniote ancestor. We identify a pleiotropic substitution that is necessary for A9 activation of TLR4, sufficient to increase TLR4 activation by the A9 amniote ancestor and played a role in loss of A9 proteolytic resistance. Mutating this site has minimal effect on A8/A9 antimicrobial activity or proteolytic resistance. Lastly, we show that proteolysis is not required for A9 activation of TLR4. Taken together, this work reveals that mammals concomitantly evolved A8/A9 antimicrobial activity, A9 proinflammatory activity, and a way to selectively regulate A9 inflammation via loss of A9 proteolytic resistance. These findings provide unprecedented mechanistic and evolutionary insight into A9 function and show how a single mutation can have pleiotropic effects in one functional state of a protein while not impacting another, thus facilitating the evolution of multifunctionality.

## Results

We first set out to establish when A9 evolved three innate immune properties: antimicrobial activity via formation of the A8/A9 complex, proinflammatory activation of TLR4 by A9 alone, and the differential proteolytic susceptibility of A9 and A8/A9.

### A9s evolved to form antimicrobial A8/A9 complexes early in mammals

We sought to determine when A9 evolved to form the antimicrobial A8/A9 complex. We hypothesized that A8/A9 antimicrobial activity evolved in the ancestor of therian mammals (the shared ancestor of marsupials and placental mammals) for several reasons. First, the broad-spectrum antimicrobial activity of human and mouse A8/A9 is well established (*Hadley et al., 2018*; *Damo et al., 2013*; *Clark et al., 2016*; *Besold et al., 2018b*; *Nakashige et al., 2016*; *Hayden et al., 2013*; *Nakashige et al., 2015*; *Liu et al., 2012*; *Brunjes Brophy et al., 2013*; *Brophy et al., 2012*; *Baker et al., 2017*; *Gagnon et al., 2015*; *Nisapakultorn et al., 2001*). Second, A9 and A8 genes are only found together in therian mammals (*Figure 1b*; *Loes et al., 2018*); therefore the A8/A9 complex could not have arisen earlier than in the ancestor of therian mammals. Lastly, the residues composing the antimicrobial hexahistidine metal-binding site are fully conserved across therian mammals (*Figure 1—figure supplement 1*).

To determine whether the antimicrobial A8/A9 complex arose in the ancestor of therian mammals, we compared human A8/A9 to two previously uncharacterized A8/A9 complexes. We first tested the antimicrobial activity of A8/A9 from opossum, which is one of the earliest-diverging mammals relative to humans that possesses both of the S100A8 and S100A9 genes. Opossum and human A8 and opossum and human A9 have sequence identities of approximately 50%, respectively (*Figure 1—figure supplement 1*). Following previous work, (*Brophy et al., 2012*) we produced a cysteine-free variant of the complex to avoid the use of reducing agents in the antimicrobial assay. We confirmed that cysteine-free opossum A8/A9 formed a heterotetramer (46.8 ± 0.7 kDa) in the presence of calcium – like the human and mouse proteins (*Streicher et al., 2010*) – using size exclusion chromatography coupled with multi-angle laser light scattering (SEC MALS, *Figure 1—figure supplement 2*).

We measured cysteine-free opossum A8/A9 antimicrobial activity against a representative gram-negative bacterium, *Staphylococcus epidermidis*. In studies of A8/A9 from other species, activity against *S. epidermidis* tracked with the broad-spectrum antimicrobial activity of the complex (*Hadley et al., 2018*). We assayed activity using a previously established in vitro antimicrobial assay that monitors bacterial growth in the absence or presence of S100 proteins (*Figure 1c*, *Figure 1—figure supplement 3*; *Brophy et al., 2012*). To compare the activity of different proteins, we quantified inhibition at seven hours (*Figure 1d*). We observed a dose-dependent decrease in *S. epidermidis* growth in the presence of low micromolar concentrations of cysteine-free opossum A8/A9 (*Figure 1c–d*, *Figure 1—figure supplement 3*). The antimicrobial activity of opossum A8/A9 was weaker than that of human A8/A9: opossum A8/A9 delayed bacterial growth, while human A8/A9 both delayed growth and decreased bacterial carrying capacity (*Figure 1c*). It was previously found that cysteine-free human A8/A9 was potently antimicrobial, (*Brophy et al., 2012*) while cysteine-free mouse A8/A9 exhibits weaker antimicrobial activity than wildtype mouse A8/A9 (*Hadley et al., 2018*). To determine whether the weaker activity of opossum A8/A9 was due to the removal of cysteines, we also measured the activity of wildtype opossum A8/A9 against *S. epidermidis*. We found that wildtype opossum A8/A9 had higher activity than cysteine-free opossum A8/A9 over the concentration range tested (*Figure 1—figure supplement 3*). This suggests that the cysteines present in mouse and opossum A8/A9 play a role in their antimicrobial activity, unlike in human A8/A9. The antimicrobial activity of the opossum A8/A9 complex thus appears to be more similar to that of mouse A8/A9 than human A8/A9.

The shared antimicrobial activity of human, mouse, and opossum A8/A9 strongly suggests that the antimicrobial A8/A9 complex evolved in the ancestor of therian mammals. To test this further, we measured the antimicrobial activity of ancestrally reconstructed therian mammalian A8/A9 (ancA8/A9 – *Figure 1b*). We used our previously published phylogenetic tree (*Loes et al., 2018*) consisting of 172 S100 sequences to reconstruct therian mammalian ancestral A8 and A9 (ancA9 and ancA8 – *Figure 1*, *supplementary files 1–2*), which were used to form the ancA8/A9 complex. AncA8 and ancA9 had average posterior probabilities of 0.88 and 0.83, with sequence similarities to human A8 and A9 of 66% and 64%, respectively (*Figure 1—figure supplement 4*). Average posterior probabilities in this range have been previously described as medium confidence reconstructions, with reconstructions characterized by others having average posterior probabilities as low as 0.7 (*Eick et al., 2016*). We confirmed that each protein was folded and had secondary structure content similar to that of human A8/A9 using far-UV circular dichroism (CD) spectroscopy (*Figure 1—figure supplement 5*).

We then measured the antimicrobial activity of ancA8/A9 against *S. epidermidis.* We observed a potent reduction in *S. epidermidis* growth comparable to that of human A8/A9 (*Figure 1c–d*, *Figure 1—figure supplement 3*). To test for the robustness of this finding to phylogenetic uncertainty, we also tested the antimicrobial activity of an AltAll (*Eick et al., 2016*) reconstruction of ancA8/A9 against *S. epidermidis* (altancA8/A9, *Figure 1c–d*, *Figure 1—figure supplement 4*). In this reconstruction, we swapped all ambiguously reconstructed amino acid positions for their second-most likely state (see methods). AncA8/A9 and altancA8/A9 differ by 27 amino acids total (10 between ancA8 and altancA8 and 17 between ancA9 and altancA9 - *Figure 1—figure supplement 4*). AltancA8/A9 exhibited antimicrobial activity against *S. epidermidis* similar to opossum A8/A9: it delayed growth but did not ultimately limit bacterial carrying capacity. While the hexahistidine site residues are conserved in ancA8/A9 and altancA8/A9 (*Figure 1—figure supplement 1*), it appears that a subset of the ambiguously reconstructed 27 residues are important for A8/A9 antimicrobial activity, perhaps affecting the orientation and/or affinity of the hexahistidine metal-binding site.

Taken together, the antimicrobial activity of modern mammalian A8/A9 complexes (human, mouse, and opossum) and the antimicrobial activity of the reconstructed ancA8/A9 complex suggest that A9s evolved to form the antimicrobial A8/A9 complex in the ancestor of mammals.

## A9s evolved potent proinflammatory activity from a weakly active amniote ancestor

We next sought to determine when A9s evolved potent proinflammatory activity via activation of TLR4. Our previous work revealed that human A9 potently activates not only human TLR4 in functional assays, but also opossum and chicken TLR4 (*Figure 2a*; *Loes et al., 2018*). In contrast, chicken MRP126, the sauropsid ortholog of A9s, was found to be a weak activator of all TLR4s, including

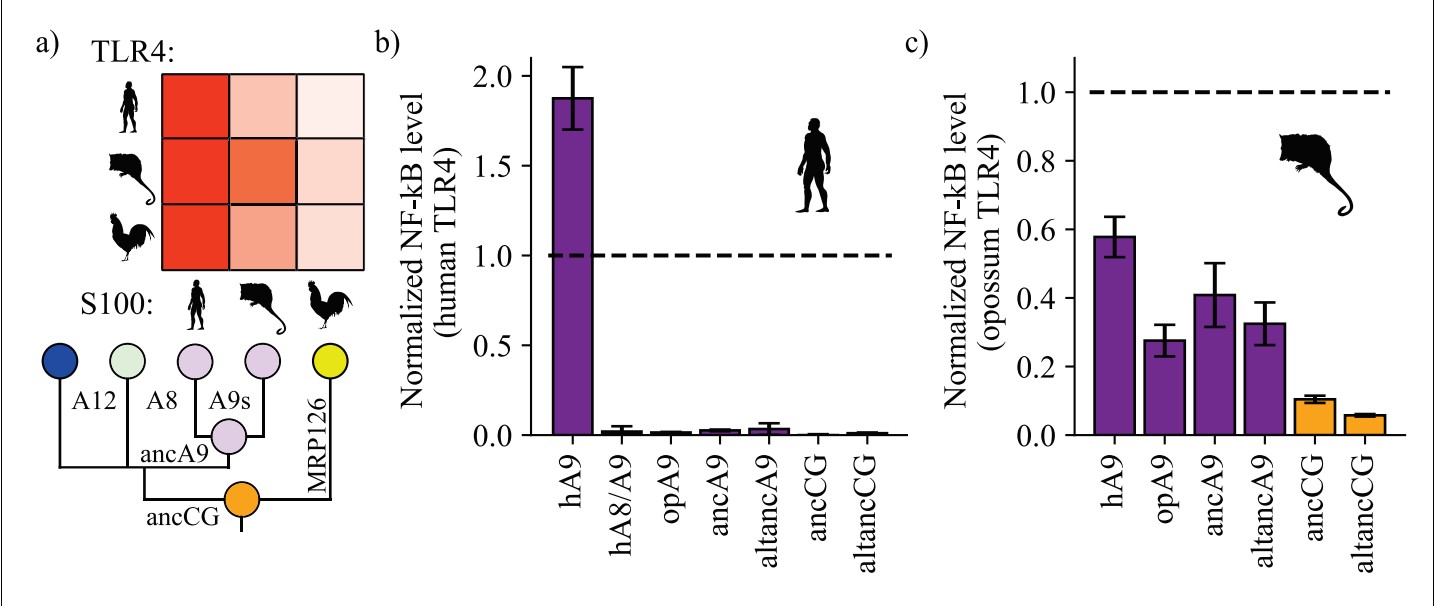

**Figure 2.** A9s gained proinflammatory activity from a weakly proinflammatory ancestor. (a) Schematic of previously measured proinflammatory activity of S100s against various TLR4s. Species labels on x and y-axes of heatmap are the same as *Figure 1*. Heatmap coloring is scaled to match 2 μM S100 activity levels measured in supplementary figure S2 of *Loes et al. (2018)*. (b) and (c) NF-κB production of human and opossum TLR4 in response to treatment with modern and ancestral S100 proteins. Bars represent average of >3 biological replicates, error bars are standard error of the mean. All values are background-subtracted and normalized to LPS positive control (see methods).

The online version of this article includes the following source data and figure supplement(s) for figure 2:

**Source data 1.** Human TLR4 activation data.
**Source data 2.** Opossum TLR4 activation data.
**Figure supplement 1.** Analysis of S100 protein LPS contamination.
**Figure supplement 2.** Dose curves for S100 activation of human TLR4.
**Figure supplement 3.** Dose curves for S100 activation of opossum TLR4.

chicken TLR4. Both human and opossum A9 activate chicken TLR4 better than chicken MRP126 does. Two possibilities are consistent with these observations. Either mammalian A9s evolved enhanced proinflammatory activity from a less active amniote ancestral state, or A9s maintained a potent ancestral activity that was lost by chicken MRP126.

To differentiate between these two possibilities, we determined the ancestral proinflammatory activity of these proteins. We used ASR to reconstruct the shared amniote ancestor of A9s, A8s, A12s, and MRP126s. This group of proteins is known collectively as the 'calgranulins', so we will refer to this ancestral protein as ancCG (ancestor of calgranulins). We also constructed an alternate, 'alt All' version of this ancestor (altancCG, *supplementary file 1*), which differed from ancCG by eight amino acids. The average posterior probability of ancCG was 0.86 (*Figure 1—figure supplement 4*). We also expressed and purified ancA9 and altancA9 – the A9 subunits from the ancestral A8/A9 complexes described above. We confirmed that each protein was folded and had secondary structure content similar to that of modern S100s using far-UV CD spectroscopy (*Figure 1—figure supplement 5*).

We then tested modern and ancestral S100s for activity against human TLR4. Following previous work, (*Vogl et al., 2007*; *Loes et al., 2018*; *Anderson et al., 2019*; *Ehrhardt et al., 2006*) we transiently transfected HEK293T cells with plasmids encoding TLR4 and its species-matched cofactors MD-2 and CD14, added purified S100 proteins to the growth media, and then measured output of luciferase under control of an NF-κB promoter. Consistent with previous results, (*Loes et al., 2018*) we found that human A9 potently activated human TLR4, resulting in high levels of NF-κB production (*Figure 2b*, *Figure 2—figure supplements 1–2*). Human A8/A9 and opossum A9 exhibited much weaker activity against human TLR4. Lastly, we tested ancA9, altancA9, ancCG, and altancCG for activity against human TLR4 and observed weak or no activation for each ancestral protein. This

result is unsurprising, as we previously found that human TLR4 is more specific than other amniote TLR4s: human TLR4 is activated much more potently by human A9 than by any other S100 protein (*Figure 2a–b*; *Loes et al., 2018*). In contrast, TLR4s from other species (mouse, opossum, and chicken) appear to be more promiscuous and can be activated similarly by S100s from various species (*Figure 2a*; *Loes et al., 2018*). This is consistent with lineage-specific coevolution between human TLR4 and human A9 – a confounding variable that makes assessment of ancestral S100 protein proinflammatory activity difficult using human TLR4.

We predicted that opossum TLR4 would be a better protein to probe ancestral S100 proinflammatory function because opossum TLR4 is broadly activated by A9s across mammals and gives little indication of lineage-specific coevolution (*Loes et al., 2018*). We therefore tested the proinflammatory activity of ancA9, ancCG, and their corresponding alternate reconstructions against opossum TLR4. Corroborating previous results, human A9 strongly activated opossum TLR4, while opossum A9 activity was approximately half that of human A9 (*Figure 2c*, *Figure 2—figure supplements 1–3*). AncA9 and altancA9 activated opossum TLR4 to the same extent as opossum A9. AncCG and altancCG were the weakest activators of opossum TLR4, with activity approximately 25% or less than that of human A9.

These findings suggest that A9s evolved enhanced proinflammatory activity early in mammals from a weakly proinflammatory amniote ancestor, while A8/A9s and chicken MRP126 maintained weak, ancestral proinflammatory activity.

## A9s evolved proteolytic susceptibility from a proteolytically resistant amniote ancestor

We next sought to determine when the differential proteolytic susceptibility of A9 and A8/A9 evolved. We used a simple in vitro assay to monitor S100 protein degradation over time in the presence of proteinase K, a potent non-specific serine protease (*Figure 3a*). Proteinase K was chosen both because of its low specificity and to mimic other serine proteases that A9 and A8/A9 encounter when released from neutrophils during an inflammatory response (*Henry et al., 2016*; *Stapels et al., 2015*; *Kessenbrock et al., 2011*; *Heutinck et al., 2010*; *Fu et al., 2018*). Proteolytic decay rates were estimated by fitting a single exponential decay function to the data (*Figure 3b*, *Figure 3—figure supplements 1–4*).

Human A8/A9 has been described as extremely resistant to proteases (*Nacken and Kerkhoff, 2007*); however, it has not been compared to S100 proteins besides human A8 and A9. To establish a baseline expectation for S100 protein proteolytic resistance, we characterized the proteolytic resistance of a broad set of human S100s against proteinase K. As previously shown, (*Nacken and Kerkhoff, 2007*) human A9 and A8 alone were rapidly proteolytically degraded, while the human A8/A9 complex exhibited strong resistance (*Figure 3c*, *Figure 3—figure supplements 1–2*). Under our conditions, the degradation rates for human A8 and A9 were approximately three orders of magnitude faster than that of the human A8/A9 heterocomplex. We then characterized closely related protein human S100A12 (A12), the chicken ortholog MRP126, and six distantly related human S100s (*Wheeler et al., 2016*). Human A12, chicken MRP126, and five out of six more distantly related human S100s exhibited intermediate to strong proteolytic resistance, each degrading 1–2 orders of magnitude slower than human A8 or A9 but, on average, one order of magnitude faster than human A8/A9 (*Figure 3c*, *Figure 3—figure supplements 1–2*). Notably, human A12 and chicken MRP126 formed predominantly homodimers by SEC MALS under these conditions (*Figure 1—figure supplement 2*), indicating that higher-order oligomerization (>2 subunits) isn't required for S100 proteolytic resistance. Lastly, human A14 degraded faster than A9 or A8. This protein is evolutionarily distant (*Wheeler et al., 2016*) and therefore likely reflects independent evolution of this property. Taken together, these data show that the A8/A9 complex, A9, and A8 indeed fall at the extremes of human S100 proteolytic resistance; human A9 and A8 are among the fastest-degrading S100s tested, while human A8/A9 is one of the slowest.

To test whether A9 and A8 proteolytic susceptibility and A8/A9 resistance are conserved across mammals, we characterized mouse and opossum A9, A8, and A8/A9 for proteolytic resistance. Mouse A9 and A8 were found to be highly proteolytically susceptible and mouse A8/A9 strongly proteolytically resistant, matching the pattern observed for their human counterparts (*Figure 3c* and *Figure 3—figure supplement 2*). Opossum A9 and A8 were also highly proteolytically susceptible, while opossum A8/A9 was resistant (*Figure 3c*, *Figure 3—figure supplement 2*). This indicates that

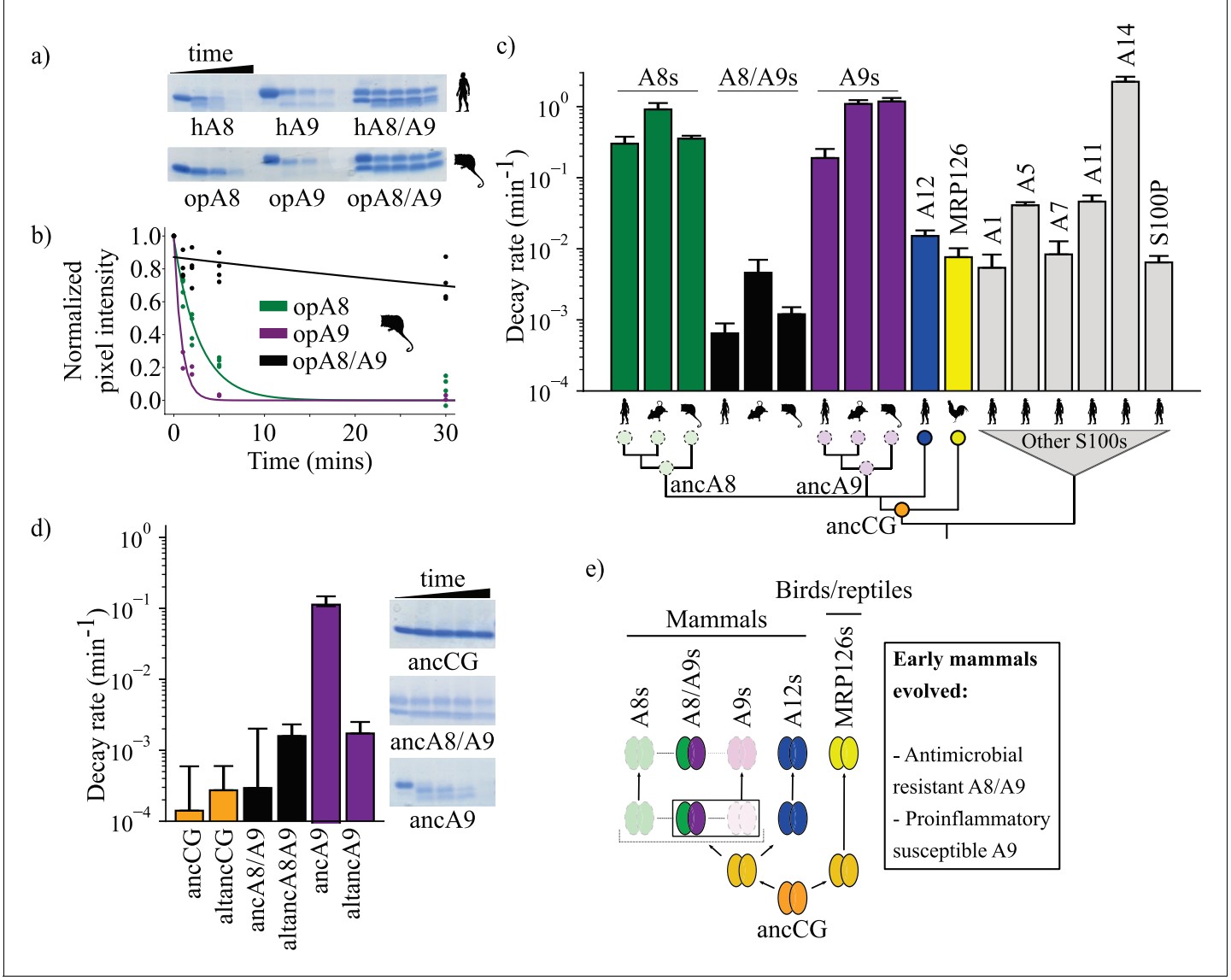

**Figure 3.** A9s lost proteolytic resistance from a proteolytically resistant amniote ancestor. (**a**) In vitro proteolytic resistance assay showing SDS-PAGE gel of S100 protein degradation via proteinase K over time. Gels were quantified using densitometry and normalized to the undigested protein band intensity. (**b**) A single exponential decay model was globally fit to the data to quantify decay rates. Points are biological replicates, lines are model fit to data. (**c**) S100 protein proteolysis rates mapped onto schematized S100 phylogeny. X-axis cartoon labels same as in *Figure 1*. Circles indicate proteolytic susceptibility (faded/dashed) and resistance (solid), with predicted resistance shown for ancA8, ancA9, and ancCG nodes. (**d**) Decay rates for ancestrally reconstructed proteins, with gels shown on the right. For panels (**c**) and (**d**), error bars are the square root of the diagonalized covariance matrix from the fit and the y-axis is in log scale. (**e**) Summary model for proposed evolution of A9 and A8/A9 innate immune properties. Box around A8/A9 and A9 indicate location in tree (ancestor of therian mammals) where immune functions evolved.

The online version of this article includes the following source data and figure supplement(s) for figure 3:

**Source data 1.** Proteolytic susceptibility data.
**Figure supplement 1.** Survey of proteolytic susceptibility across modern S100 proteins.
**Figure supplement 2.** Comparison of proteolytic susceptibility for A8s, A9s, and A8/A9 complexes across mammals.
**Figure supplement 3.** Comparison of proteolytic susceptibility for ancestrally reconstructed S100s.
**Figure supplement 4.** Comparison of proteolytic S100 protein mutants at position 63.

the susceptibility of A9s and A8s and the resistance of A8/A9 complexes is conserved across mammals.

When mapped onto the S100 phylogeny, the most parsimonious explanation for these data is that the shared amniote ancestor—ancCG—was proteolytically resistant (*Figure 3c*). In this scenario,

A12s, MRP126s, and A8/A9s conserved ancestral resistance, while A9s and A8s independently lost resistance early in mammals. Alternatively, ancCG could have been proteolytically susceptible. This would mean that A9s and A8s maintained an ancestral susceptibility, while MRP126s, A12s, and A8/A9s each evolved novel proteolytic resistance.

To distinguish between these possibilities, we characterized ancestrally reconstructed S100s for proteolytic resistance. AncCG and altancCG exhibited extremely high proteolytic resistance (*Figure 3d*, *Figure 3—figure supplement 3*), with degradation rates 3–4 orders of magnitude slower than modern A8s or A9s and approximately one order of magnitude slower than modern A8/A9s. AncA8/A9 and altancA8/A9 also demonstrated high proteolytic resistance, with degradation rates approximately 2–3 orders of magnitude slower than A8s and A9s and comparable to modern A8/A9 complexes (*Figure 3*). Together, these data paint a consistent picture: the amniote ancestor of A9s, ancCG, was strongly resistant to proteolytic degradation. Modern A9s and A8s lost proteolytic resistance from an ancestrally resistant state, while modern A12s, A8/A9 complexes, and MRP126s maintained the ancestral proteolytic resistance (*Figure 3e*).

Finally, we sought to better resolve when A9s acquired proteolytic susceptibility. We hypothesized that this occurred in the ancestor of mammalian A9s before the divergence of therian mammals and marsupials. To test this hypothesis, we measured the proteolytic susceptibility of therian mammalian ancA9 and found that it degraded rapidly (*Figure 3*). However, its alternative reconstruction (altancA9), was slow to degrade, with a rate two orders of magnitude slower than ancA9 and comparable to other highly resistant S100s. Because the descendants of ancA9 all exhibit proteolytic susceptibility (*Figure 3*), the simplest explanation is that altancA9 is a low-quality reconstruction that does not capture the properties of the historical protein. Alternatively, proteolytic susceptibility could have been independently acquired along marsupial and placental mammal lineages.

## A single substitution had pleiotropic effects on A9 proinflammatory activity and proteolytic resistance

We found above that A9 evolved to form the antimicrobial A8/A9 complex, gained potent proinflammatory activity, and lost proteolytic resistance over the narrow evolutionary interval after the divergence of mammals and sauropsids but before the divergence of placental mammals and marsupials. We next sought to determine how A9 evolved its antimicrobial and proinflammatory activities and lost proteolytic resistance.

The mechanism by which A9 evolved to form the antimicrobial A8/A9 complex is straightforward. After ancCG duplicated, additional histidines accumulated in the mammalian A8 and A9 ancestors that created the antimicrobial hexahistidine metal-binding site in the A8/A9 complex (*Figure 3e* and *Figure 1—figure supplement 1*). A8s acquired one additional histidine while retaining the three histidines present in ancCG, while A9s acquired two additional histidines via acquisition of a C-terminal extension (*Figure 1—figure supplement 1*). While A9s evolved five of the six histidines composing the hexahistidine metal-binding site, this was not sufficient to convey potent antimicrobial activity (*Figure 1c*). Instead, preservation of A8/A9 heterocomplex formation resulted in proper assembly of the complete antimicrobial hexahistidine site early in mammals. The quantitative difference between ancA8/A9 and altancA8/A9 antimicrobial activity suggests that other amino acid changes tuned the antimicrobial activity of the molecule, but the core functionality is determined by whether the six histidine residues were present. This is independently supported by Brunjes Brophy et al., who showed that mutating the two C-terminal histidines in A9 is sufficient to strongly decrease the A8/A9 complex's antimicrobial activity (*Brunjes Brophy et al., 2013*).

The mechanisms by which A9s gained proinflammatory activity and lost proteolytic resistance are less obvious, particularly because the mechanism by which A9 activates TLR4 is not well understood. We reasoned that we could identify functionally important amino acid substitutions by focusing on the evolutionary interval over which these properties evolved. We therefore compared the sequences of ancCG (weakly proinflammatory and resistant to proteolytic degradation) and ancA9 (potently proinflammatory and susceptible to proteolytic degradation). We further narrowed down sequence changes of interest by looking for residues conserved in modern A9s (*Figure 4—figure supplement 1*). Finally, we focused on amino acid changes in helix III of A9, as this region is thought to be important for A9 activation of TLR4 based on in vitro binding studies and in silico docking studies (*Vogl et al., 2018*). Only one historical amino acid substitution met all three criteria: position 63 (human A9 numbering). This residue is a phenylalanine in both ancCG and altancCG, is conserved as

a phenylalanine in 95% of modern A8s and A12s and has been substituted for a methionine or leucine (M/L) in 97% of A9s (*Figure 4a*).

We hypothesized that reverting this site to its amniote ancestral state—M63F—might affect A9 proinflammatory activity. We mutated this position to a phenylalanine in human A9 and opossum A9 and tested each protein for TLR4 activation. Strikingly, we found that introducing M63F into human A9 severely compromised its ability to activate human TLR4 (*Figure 4b*, *Figure 2—figure supplement 2*). This was also true for opossum A9: introduction of M63F (human numbering) strongly decreased opossum A9 activation of opossum TLR4 (*Figure 4c*, *Figure 2—figure supplement 3*). We next introduced the forward substitution, F63M, into ancCG and tested its proinflammatory activity against opossum TLR4. We observed a modest increase in ancCG activity with the F63M substitution, with activity comparable to that of opossum A9 (*Figure 4c*, *Figure 2—figure supplement 3*).

For most proteins we studied, the amino acid at position 63 did indeed play an important role in determining the pro-inflammatory activity of A9. The effects of toggling position 63 between Met and Phe were not, however, universal. We introduced M63F into ancA9 and observed no change in proinflammatory activity (*Figure 4c*, *Figure 2—figure supplement 3*). Further, altancA9 has a Phe at position 63 but activates TLR4 in the assay (*Figure 2c*, *Figure 2—figure supplement 3*). Thus, while position 63 is an important contributor to activity in modern A9s, other substitutions were also important for the transition from a weakly pro-inflammatory ancestor to the modern set of potently pro-inflammatory A9s.

Because A9s lost proteolytic resistance and gained proinflammatory activity over the same evolutionary time interval, we reasoned that the F63M substitution might have also played a role in A9 loss of proteolytic resistance. To test this, we characterized the proteolytic resistance of human A9 M63F and ancA9 M63F. Strikingly, reversion of this single mutation rendered both ancA9 and human A9 strongly resistant to proteolytic degradation, decreasing their respective degradation rates by 1–2 orders of magnitude and approaching the degradation rates of ancCG and various A8/A9 complexes (*Figure 4d*, *Figure 3—figure supplement 4*). To relate these findings to proteases that A9 might encounter at sites of inflammation, we also measured the proteolytic resistance of human A9 and human A9 M63F against two neutrophil-specific proteases – cathepsin G and neutrophil elastase (*Figure 4—figure supplement 2*). Neutrophils release these proteases along with A9 at sites of inflammation, often through Neutrophil Extracellular Traps (NETs) (*Henry et al., 2016*; *Stapels et al., 2015*; *Kessenbrock et al., 2011*; *Heutinck et al., 2010*; *Janoff, 1972*; *Jerke et al., 2015*; *O'Donoghue et al., 2013*). We found that M63F decreased the rate of human A9 degradation in the presence of cathepsin G and neutrophil elastase in vitro by approximately one order of magnitude, matching our results using proteinase K (*Figure 4—figure supplement 2*). Lastly, we tested the effect of the forward mutation – F63M – on ancCG proteolytic resistance. We observed no change in resistance for ancCG F63M, indicating that additional substitutions were required to render ancA9 proteolytically susceptible. Together these data show that a single historical reversion is sufficient to render A9s proteolytically resistant, indicating that this position played a role in the loss of A9 proteolytic resistance early in therian mammals.

## The pleiotropic substitution minimally affects the A8/A9 complex

A primary goal of this study was to understand the role of pleiotropy in the evolution of multifunctionality. M63F clearly has pleiotropic effects on A9, altering both its proinflammatory activity and proteolytic resistance (*Figure 4b–d*). We next asked whether introducing M63F would pleiotropically affect the antimicrobial A8/A9 complex. Position 63 is somewhat distant from the A8/A9 interface and the antimicrobial hexahistidine site (~10 Å in the manganese-bound A8/A9 crystal structure) (*Damo et al., 2013*); we therefore hypothesized that M63F should not affect A8/A9 complex formation or function. To test this hypothesis, we introduced M63F into human A8/A9 and tested it for oligomeric state, proteolytic resistance, and antimicrobial activity against *S. epidermidis*. As predicted, human A8/A9 M63F predominantly formed a heterotetramer in the presence of calcium by SECMALS with a molecular weight similar to that of wildtype human A8/A9 (48.7 ± 4.2 kDa – *Figure 1—figure supplement 2*). We found that human A8/A9 M63F was also strongly resistant to proteolytic degradation, similar to human A8/A9 (*Figure 4d*). Lastly, M63F had minimal impact on human A8/A9 antimicrobial activity against *S. epidermidis*, retaining potent antimicrobial activity (*Figure 4e*). In contrast, neither human A9 nor human A9 M63F were antimicrobial against *S.*

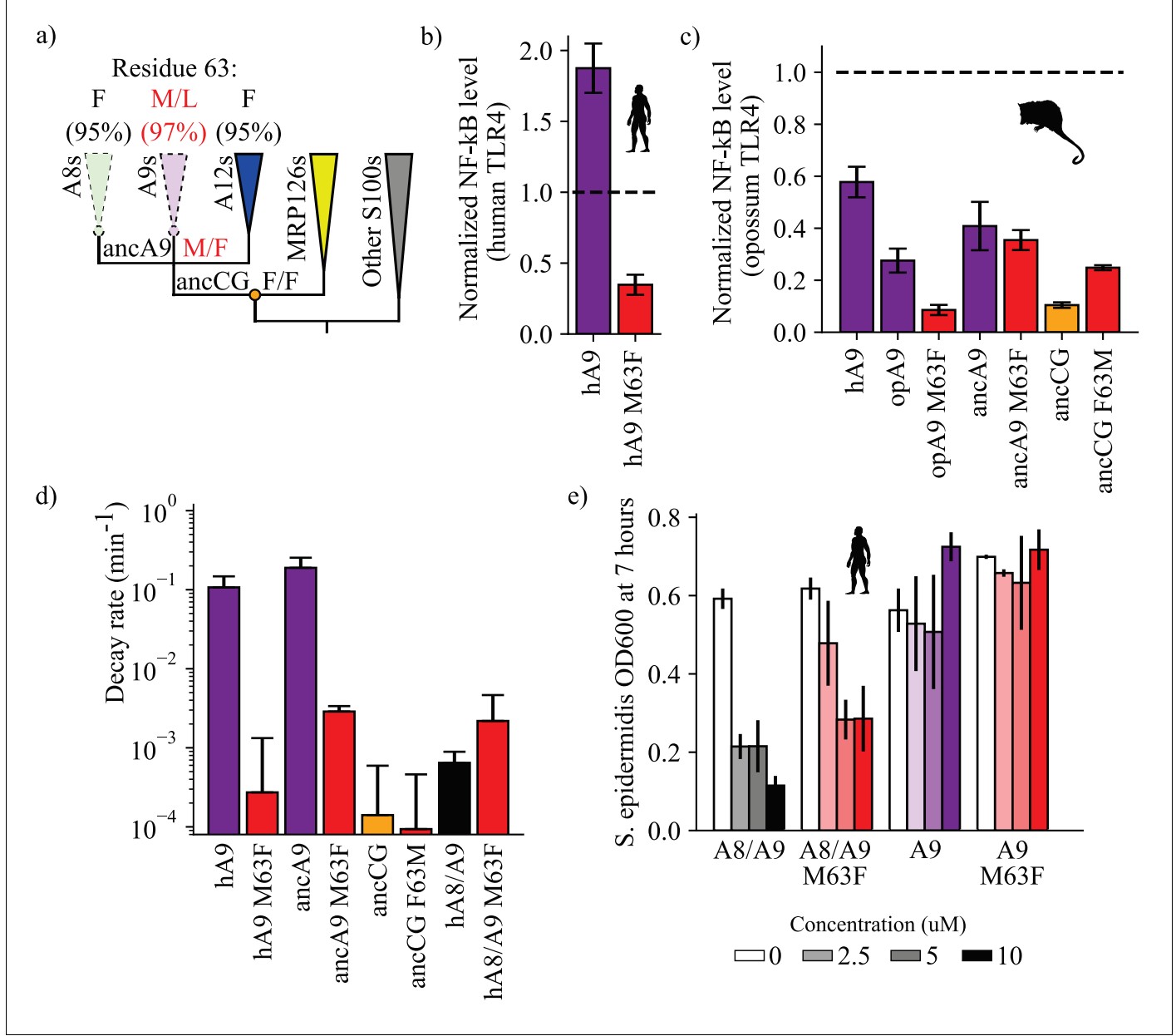

**Figure 4.** A single historical substitution affects A9 proinflammatory activity and proteolytic resistance without affecting A8/A9 proteolytic resistance or antimicrobial activity. (**a**) Schematic S100 phylogenetic tree with the amino acid state of position 63 shown at key nodes. Wedges represent clades, colored as in *Figure 1*. Lines indicate proteolytic susceptibility (faded/dashed) and resistance (solid). Circles indicate characterized ancestors. Amino acid labels represent maximum likelihood state/alternate amino acid state for position 63 at ancestral nodes, while labels at clade tips represent percent conservation across modern S100 protein sequences. (**b–c**) NF-κB production of S100 point mutants at position 63 against human (**b**) and opossum (**c**) TLR4. (**d**) Proteolysis rates for S100 point mutants at position 63 (human A9 numbering). Error bars and y-axis are the same as in *Figure 1*. (**e**) Antimicrobial activity of hA9 and hA8/A9 with and without M63F mutation against *S. epidermidis*. Axes and error bars same as in *Figure 1d*. The online version of this article includes the following source data and figure supplement(s) for figure 4:

**Source data 1.** Human TLR4 activation data with point mutants.
**Source data 2.** Proteolytic susceptibility data with point mutants.
**Source data 3.** Antimicrobial activity data.
**Source data 4.** Opossum TLR4 activation data with point mutants.
**Figure supplement 1.** Changes at position 63 correlate with A9 proinflammatory activity and proteolytic resistance.
**Figure supplement 2.** Comparison of human A9 and human A9 M63F proteolytic resistance against human proteases.

*epidermidis* (*Figure 4e*). These findings suggest that this single amino acid position had important effects on the evolution of A9 activation of TLR4 and loss of proteolytic resistance without significantly impacting A8/A9 oligomeric state, proteolytic resistance, or antimicrobial activity.

## M63F increases protein thermodynamic stability and decreases unfolding rate of human A9

We next asked what effect M63F has on the biophysical properties of human A9. Residue 63 sits in the middle of helix III of A9, pointing inward toward helix II, and is neither a core residue nor fully surface-exposed (*Figure 5a*; *Itou et al., 2002*). Based on the published structure of human A9, (*Itou et al., 2002*) a Phe at position 63 could be plausibly tolerated without a steric clash. Using circular dichroism (CD) spectroscopy, we found that the bulk secondary structure content of human A9 M63F was similar to that of hA9 (*Figure 5b*). We measured the oligomeric state of human A9 M63F by SEC MALS and found that it predominantly forms a homodimer in solution similarly to human A9, with no detectable monomers or larger oligomers (*Figure 5c and f*). These data together indicate that M63F does not significantly alter human A9's secondary structure or oligomeric state.

We then examined whether M63F alters the stability of human A9. We measured equilibrium unfolding curves for human A9 and human A9 M63F using CD spectroscopy and chemical denaturation via urea. We found that M63F appears to stabilize human A9, increasing the apparent free energy of unfolding by more than four kcal/mol and shifting the $C_m$ by ~2M urea (*Figure 5d and f*, *Figure 5—figure supplement 1*). We also measured the unfolding kinetics of human A9 and human A9 M63F in the presence of calcium by spiking protein directly into 6M guanidinium hydrochloride (gdn-HCl) denaturant and monitoring its unfolding rate by CD spectroscopy. Strikingly, human A9 M63F takes several minutes to unfold under these conditions, while human A9 unfolds immediately within the dead time of the experiment (*Figure 5e–f*, *Figure 5—figure supplement 2*). We note that the folding pathway for A9 is complex and almost certainly not two-state—calcium binding, monomer folding, and dimerization all contribute—and thus we cannot reliably determine how M63F affects the stability of each of these potential folding intermediates. The large increase in apparent stability and unfolding rate suggests, however, that the mutation stabilizes some aspect of the folded structure.

## Proteolysis is not required for A9 activation of TLR4

The work above identified a mutation that, when introduced into human A9, increases the stability of the protein while also potently compromising its ability to activate TLR4. The mutation is not at a surface position and is therefore not likely a direct participant in the A9/TLR4 protein/protein interface. Further, the same mutation dramatically decreases the proteolytic susceptibility of the protein. One simple way to explain these observations would be if the proteolytic susceptibility itself was the feature that evolved to allow activation of TLR4. This would be consistent with a previous observation that proteolytic products of A9 activate TLR4 (*Vogl et al., 2018*).

To test whether proteolysis itself was sufficient for activity, we engineered an alternate variant of A9 that was proteolytically susceptible. We introduced the M63A mutation into human A9, anticipating that the short alanine sidechain would not have the stabilizing effect of M63F. As expected, human A9 M63A was highly susceptible to proteolytic degradation, similar to wildtype human A9 (*Figure 6a*, *Figure 3—figure supplement 4*). We reasoned that if proteolysis is the primary determinant of A9 activation of TLR4, then proteolytically susceptible human A9 M63A should potently activate TLR4. Human A9 M63A, however, exhibited diminished proinflammatory activity, similar to human A9 M63F (*Figure 6b*, *Figure 2—figure supplement 2*). This indicates that the methionine at position 63 is important for A9 activation of TLR4. Further, we quantified the amount of human A9, human A9 M63F, and human A9 M63A before and after measuring TLR4 activity and observed no decrease in the amount of full-length protein remaining for wildtype human A9 or either mutant by western blot (*Figure 6c*). This indicates that A9 is not digested by extracellular proteases over the course of the ex vivo assay and that proteolysis is not necessary for A9 activation of TLR4.

Although proteolysis does not appear to be a requirement for TLR4 activation, this does not rule out that proteolysis could increase A9 proinflammatory activity by releasing proinflammatory fragments of A9. To test for this possibility, we treated human A9 with agarose-immobilized proteinase K for increasing amounts of time, removed the protease, and then measured the proinflammatory

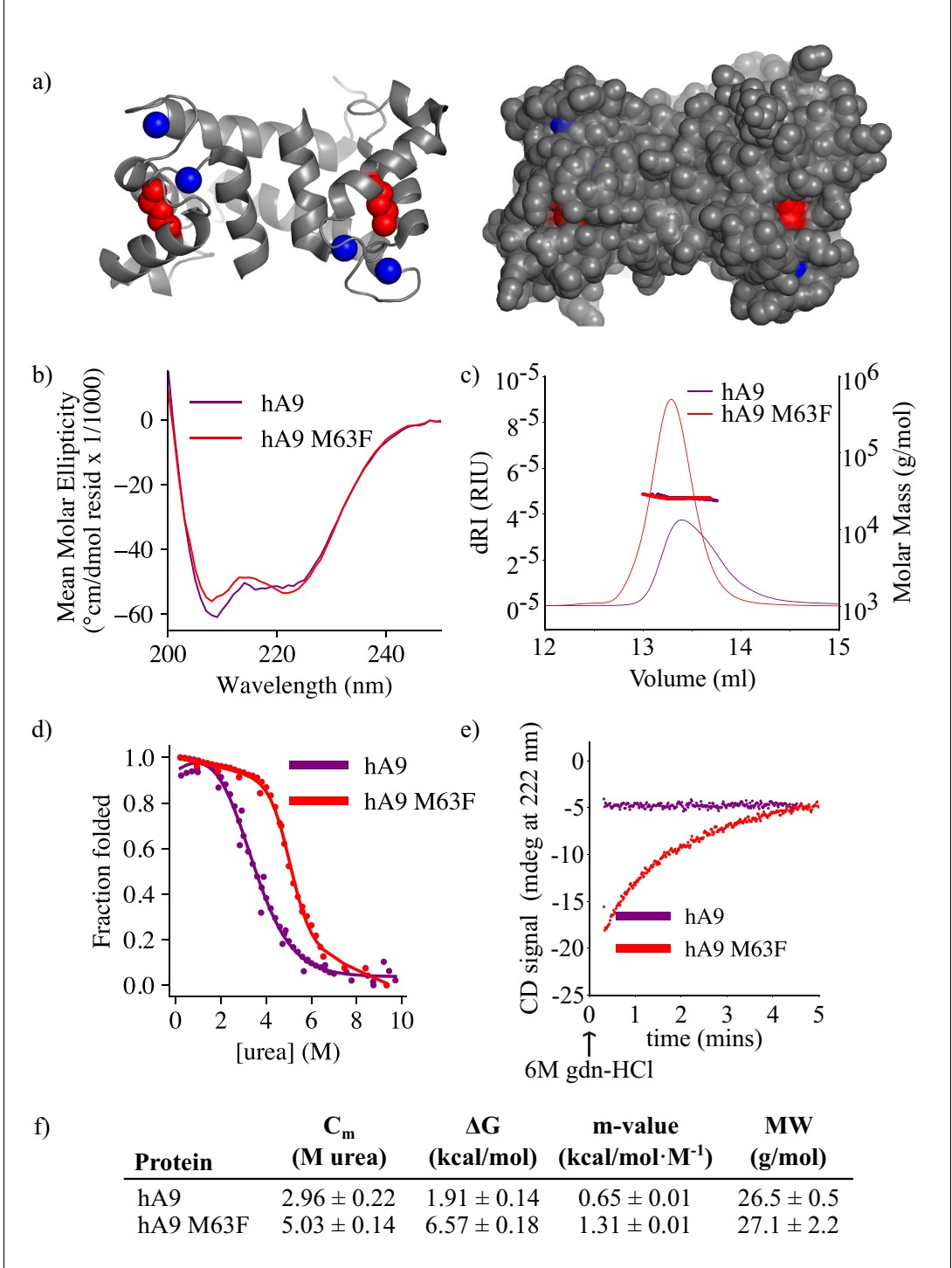

**Figure 5.** M63F increases human A9 apparent stability by decreasing its unfolding rate. (**a**) Crystal structure of hA9 (PDB entry 1irj) (*Itou et al., 2002*). Cartoon depiction left, surface view right. Calcium ions are blue spheres. M63 is highlighted in red – two total for homodimeric A9. (**b**) Far-UV circular dichroism (CD) spectroscopy scans of hA9 and hA9 M63F. Data represent average of 3 scans. (**c**) SEC MALS analysis of hA9 and hA9 M63F oligomeric state. Solid lines are refractive index (left y-axis), points and molecular weights in table below represent molar mass calculated from light scattering detectors using ASTRA software (right y-axis - see methods). (**d**) Equilibrium chemical denaturation (urea) of 5 µM hA9 and hA9 M63F monitored by CD at 222 nm. Solid lines represent two-state unfolding model fit to data. (**e**) Kinetics of hA9 and hA9 M63F unfolding via chemical denaturation (guanidinium hydrochloride). Graph depicts one representative unfolding experiment. (**f**) Thermodynamic parameters estimated from (**d**) and molecular weights estimated from (**c**). Errors are standard deviations calculated from fit (see methods).

The online version of this article includes the following source data and figure supplement(s) for figure 5:

*Figure 5 continued on next page*

*Figure 5 continued*

**Source data 1.** CD spectroscopy data.
**Source data 2.** CD unfolding kinetics data.
**Source data 3.** Chemical denaturation data by CD.
**Figure supplement 1.** Chemical denaturation of hA9 and hA9 M63F.
**Figure supplement 2.** Unfolding kinetics of hA9 and hA9 M63F.

activity of A9 degradation products (*Figure 6d*). If proteolytic products of A9 are the most proinflammatory form of the protein, we might expect to observe a spike in TLR4 activation upon A9 digestion. Instead, we observed a steady decrease in human A9 activity with increasing digestion time. This suggests that full-length human A9 is the most potent activator of TLR4.

We did observe moderate activity for proteolytic products of human A9, as previously shown (*Vogl et al., 2018*). After 30 min of digestion, no detectable full-length A9 remains by western blot (<30 ng, *Figure 6d*), but NF-κB production is still quite high, revealing that smaller fragments of A9 are sufficient to provide some degree of activation of TLR4. This raised the possibility that part of M63F's deleterious effect on proinflammatory activity could be to limit the release of active proteolytic fragments of A9. To test this, we also measured human A9 M63F activation of TLR4 after digestion for multiple hours (*Figure 6d*). Unlike wildtype, however, fragments of human A9 M63F did not activate TLR4—even after being liberated by the protease. This strongly suggests that the historical mutation induced a change in the native structure or dynamics of the molecule to bring about increased activity, independent of its effect on proteolytic susceptibility.

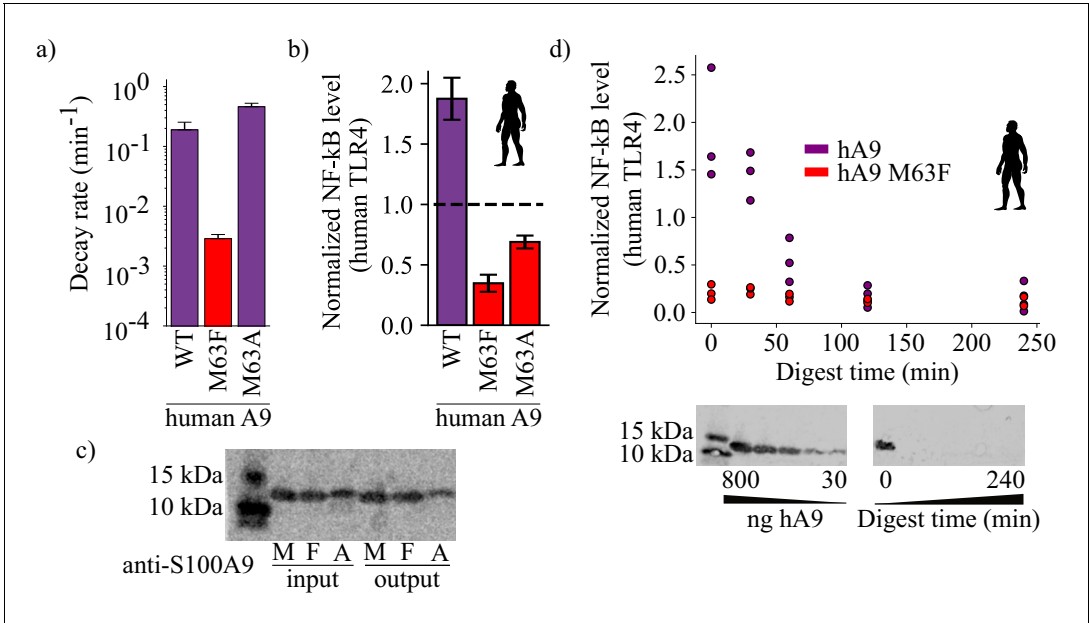

**Figure 6.** Proteolysis is not required for A9 activation of TLR4. (a) Proteolytic decay rates for human point mutants at position 63. Error bars and axes are the same as in *Figure 3*. (b) NF-κB production of human TLR4 in response to treatment with hA9, hA9 M63F, and hA9 M63A. Error bars the same as in *Figure 2*. (c) Western blot of hA9 and position 63 point mutants before and after proinflammatory activity assay. Left bands represent 10 and 15 kDa ladder. (d) NF-κB production of human TLR4 in response to hA9 and hA9 M63F pre-proteolyzed with proteinase K for increasing amounts of time. Points are biological replicates and are the average of three technical replicates. Western blots below depict the amount of full-length A9 remaining over time. Left blot shows antibody sensitivity to A9, right shows digestion time course samples. Ladder and antibody same as in (c).
The online version of this article includes the following source data for figure 6:

**Source data 1.** Proteolytic degradation data with hA9 M63A data included.
**Source data 2.** TLR4 activation data with M63A included.

## Discussion

The work presented here provides insight into how the multifunctional protein A9 evolved critical innate immune functions. We find that mammalian A9s gained enhanced proinflammatory activity and lost proteolytic resistance from a weakly proinflammatory, proteolytically resistant amniote ancestor. A single substitution played a key role in the evolution of these properties without significantly affecting the antimicrobial activity of the A8/A9 heterocomplex. This work contributes to our mechanistic understanding of how A9 activates TLR4 to drive inflammation and clarifies the role of proteolysis in A9 innate immune function.

### Innate immune functions of A9 continued to evolve within the mammals

Our data suggest that the proinflammatory and antimicrobial activities of A9 and the A8/A9 complex have undergone further optimization in placental mammals since these functions evolved. While the histidines composing the high-affinity metal-binding site of A8/A9 complexes are conserved, we observed differences in antimicrobial potency for different A8/A9 complexes. In particular, human A8/A9 is one of the most potently antimicrobial A8/A9 complexes characterized. This suggests that further optimization of the metal-binding site has occurred in along the human lineage within mammals. We also observed differences in activation of TLR4 by different A9s—human A9 is a potent, promiscuous activator of TLR4s from multiple species, while earlier-diverging A9s and other S100s exhibit weaker proinflammatory activity (*Loes et al., 2018*). Future studies are necessary to understand how, mechanistically, later-diverging A9s and A8/A9 complexes have optimized these critical innate immune functions.

### Why did A9s lose proteolytic resistance?

While proteolysis is not required for A9 activation of TLR4, it remains unclear why A9s lost proteolytic resistance. We suggest three possibilities. The first is that loss of proteolytic resistance in A9s was simply a byproduct of evolving proinflammatory activity. No A9 characterized in this study, with the exception of the alternate reconstruction of ancA9, is both proteolytically resistant and potently proinflammatory. This indicates that the molecular requirements for A9 proteolytic resistance may be incompatible with those required for A9 activation of TLR4: A9s may have gained proinflammatory activity at the expense of proteolytic resistance. A second possibility is that A9 proteolytic susceptibility is being maintained to actively remove proinflammatory A9 from the cell and retain the antimicrobial A8/A9 complex. The last possibility for A9 loss of proteolytic resistance is adaptive constraint. There could be selection for some property of A9 or A8/A9 that we did not measure that is incompatible with A9 proteolytic resistance.

While we cannot explicitly distinguish between each of these possibilities, the end result is that A9s lost proteolytic resistance from a resistant ancestor. As A9s activate TLR4 in the protease-rich extracellular space, the functional result of A9 loss of proteolytic resistance is that A9s evolved a proteolytic 'timer' concomitantly with evolving proinflammatory activity, all without affecting A8/A9 function.

### Novel mechanistic insight into A9 activation of TLR4

Our findings suggest new directions for understanding how A9 potently activates TLR4. TLR4-driven inflammation has been the focus of intense study for over 20 years, (*Källberg et al., 2012*; *Laouedj et al., 2017*; *He et al., 2016*; *Gao et al., 2015*; *Lee et al., 2016*; *Anderson et al., 2019*; *Ibrahim et al., 2013*; *Nagai et al., 2002*; *Poltorak et al., 1998*; *Prince et al., 2011*) and the structural basis of TLR4 activation by exogenous agonists, such as the bacterial cell wall component lipopolysaccharide (LPS), is well understood (*Park et al., 2009*). In contrast, little is known about how A9 activates TLR4. We have shown here that proteolytic degradation appears dispensable for activation; however, smaller fragments of the protein are sufficient activate TLR4 (*Figure 6*). Given the effect of mutating position 63 on A9 proinflammatory activity, we propose that the region surrounding it—helix III—is important for activity. This is independently supported by Vogl et al., who identified four pairs of double mutants within helix III (amino acids 64, 65, 73, and 77) that, when mutated to alanines in pairs, decrease A9 binding to TLR4 in vitro (*Vogl et al., 2018*). Biophysical characterization of hA9 M63F (*Figure 5*) indicates that it is more stable and unfolds more slowly, yet it maintains its bulk secondary structure and oligomeric state. The simplest explanation for these data is that M63F

is affecting some functionally important dynamic process of the protein, possibly mediated by helix III, that is critical for A9 activation of TLR4. The proteolytic susceptibility of A9s also supports this hypothesis, as proteolysis is a dynamic process that often relies on substrate flexibility and local unfolding events to proceed (*Guharoy et al., 2016*; *Fontana et al., 1997*; *Hubbard, 1998*; *Ottesen, 1967*; *Imoto et al., 1986*). Damage-Associated Molecular Patterns (DAMPs) often interact with their targets via hydrophobic surfaces (*Garg et al., 2010*; *Rubartelli and Lotze, 2007*; *Bianchi, 2007*); one possibility is that A9 undergoes a local unfolding event that exposes a hydrophobic surface to interact with TLR4. This would mean that studies of the native structure of A9 might not be sufficient to gain mechanistic understanding of how it activates TLR4. Further work is required to understand the nature of the active functional state of A9.

## Pleiotropic mutations can facilitate the evolution of multifunctionality

Finally, our results suggest a positive role for pleiotropy in the evolution of protein function. Pleiotropy is often viewed as a constraint on evolution: as functional complexity is added to a polypeptide sequence, it becomes increasingly challenging to introduce substitutions—and new functions—without perturbing existing ones (*Lee et al., 2018*; *Stearns, 2010*; *Paaby and Rockman, 2013*; *He and Zhang, 2006*; *Pavličev and Cheverud, 2015*; *Hirano, 1999*). Here, however, we find a single mutation that had beneficial pleiotropic effects on two important properties of A9: proinflammatory activity and proteolytic susceptibility. If A9 evolved potent proinflammatory activity without gaining susceptibility, it could potentially overstimulate inflammation simply by lingering in the extracellular milieu. Since both properties evolved at once, however, mammals evolved a proinflammatory molecule with a built-in 'timer': they gained a new inflammatory signal while avoiding potentially deleterious effects. This shows how pleiotropy can positively contribute to the evolution of new functions.

This same mutation, in contrast, had little pleiotropic effect on another functional state of A9: the A8/A9 complex. The antimicrobial activity of the A8/A9 complex was insulated from any pleiotropic effects from the mutation because proinflammatory and antimicrobial activities were partitioned between A9 and the A8/A9 complex, respectively. A mutation arose in the A9 amino acid sequence and is thus present in both A9 and A8/A9 states, but we only observe effects on the A9 state. This shows that pleiotropic constraint can be reduced when protein functions are partitioned amongst different protein states.

These findings reveal the diversity of pleiotropic roles that a single mutation can play. It further shows how the deleterious pleiotropic effects of mutations can be reduced by partitioning protein functions and properties into different functional states, thus enabling the acquisition, optimization, and expansion of new protein functions. Given the vast diversity of protein functional domains and protein-protein interactions in biology, we suspect that this is a common occurrence in the evolution of protein multifunctionality.

## Materials and methods

### Key resources table

| Reagent type (species) or resource | Designation | Source or reference | Identifiers | Additional information |
|---|---|---|---|---|
| Software, algorithm | PAML4 | *Yang, 2007* | | For ancestral sequence reconstruction |
| Software, algorithm | PhyML | *Guindon et al., 2010* | | Published tree details in *Loes et al., 2018* |
| Recombinant DNA reagent | pETDuet-1 (plasmid) | Millipore | 71146–3 | For S100 protein recombinant expression |
| Recombinant DNA reagent | pET His6 MBP TEV LIC (plasmid) | Addgene | 29656 | For S100 protein recombinant expression |
| Strain, strain background (*Escherichia coli*) | XL10-Gold Ultracompetent Cells | Agilent Technologies | 200315 | For plasmid storage/propagation |

*Continued on next page*

*Continued*

| Reagent type (species) or resource | Designation | Source or reference | Identifiers | Additional information |
|---|---|---|---|---|
| Strain, strain background (*Escherichia coli*) | Rosetta(DE3)pLysS Competent Cells | Millipore | 70956–3 | For recombinant protein expression |
| Commercial assay, kit | QuickChange Lightning Kit | Agilent Technologies | 210519 | For mutagenesis |
| Peptide, recombinant protein | Proteinase K from *Tritirachium album* | Sigma Aldrich | P2308 | |
| Peptide, recombinant protein | Cathepsin G from human neutrophils | Athens research | 16-14-030107 | |
| Peptide, recombinant protein | Neutrophil elastase from human neutrophils | Millipore Sigma | 324681 | |
| Software, algorithm | Gelquant | This paper | | Python-based scripts for gel densitometry analysis |
| Cell line (*Homo sapiens*) | HEK293T/17 | ATCC CRL-11268 | | For transient transfections |
| Transfected construct (human) | Human TLR4 | Addgene | #13086 | |
| Transfected construct (human) | Human CD14 | Addgene | #13654 | |
| Transfected construct (human) | ELAM-Luc | Addgene | #13029 | |
| Transfected construct (human) | Human MD2 | DNASU repository | HsCD00439889 | |
| Transfected construct (human) | Opossum TLR4 | Genscript | *Loes et al., 2018* (UniProt #F6Y6W8) | In pcDNA3.1(+) backbone |
| Transfected construct (human) | Opossum MD2 | Genscript | *Loes et al., 2018* (UniProt #F6QBE6) | In pcDNA3.1(+) backbone |
| Transfected construct (human) | Opossum CD14 | Genscript | *Loes et al., 2018* (NCB Accession # XP_007473804.1) | In pcDNA3.1(+) backbone |
| Biological sample (*Escherichia coli*) | K-12 LPS | Invivogen | tlrl-eklps | |
| Commercial assay, kit | Dual-Glo luciferase assay system | Promega | E2940 | |
| Chemical compound, drug | Polymyxin B | Sigma Aldrich | P4932-1MU | |
| Peptide, recombinant protein | Agarose-immobilized proteinase K | Sigma Aldrich | P9290-10UN | |
| Antibody (mouse monoclonal) | Anti-S100A9 M13 Clone 1CD22 (monoclonal) | Abnova | H00006280-M13 | WB 1:1000 |
| Antibody (goat polyclonal) | IRDye Goat anti-mouse 800CW IgG (H+L) | Licor | 926–32210 | WB 1:10,000 |

## Phylogenetics and ancestral sequence reconstruction

We reconstructed ancestral sequences using a previously published a phylogenetic tree of S100 proteins containing 172 sequences from 30 amniote taxa (*Supplementary files 1–2*; *Loes et al., 2018*). We used PAML4 to generate maximum likelihood ancestors (marginal probability method) (*Yang et al., 1995*; *Yang, 2007*) using the previously-identified maximum likelihood (ML) substitution model (LG+$\Gamma_8$) (*Jones et al., 1992*) on the ML tree. To account for reconstruction uncertainty, we also generated 'altAll' versions of each ancestor (*Eick et al., 2016*). We took every site in which the alternate reconstruction had a posterior probability >0.20 and substituted that amino acid into the maximum-likelihood ancestor. These alternate reconstructions had an average of 12 sequence differences relative to the maximum-likelihood ancestors (*Figure 1—figure supplement 4*). They represent a 'worst case' reconstruction relative to our best, maximum likelihood reconstruction.

We also investigated the effect of topological uncertainty on our reconstructed ancestors. In the published phylogenetic analysis, A8s, A9s, A12s, and MRP126s all formed distinct and well-supported clades; however, the branching pattern between these four clades could not be resolved with high confidence (*Loes et al., 2018*). To explore how this uncertainty altered our reconstructed ancestral proteins, we constructed all 15 possible topologies for the A8, A9, A12, and MRP126 clades—i.e ((A8,A9),(A12,MRP126)), ((A8,A12),(MRP126,A9)), etc.—while maintaining species-corrected, within-clade topologies. We then optimized the tree branch lengths and substitution rates for each tree using PhyML (*Guindon et al., 2010*). Finally, we used PAML to reconstruct ancA9, ancCG, and ancA8 for all 15 possible arrangements of the MRP126, A12, A8, and A9 clades. The average number of sequence differences for ancestors reconstructed using different topologies was less than or equal to the number of sequence differences between the ML and altAll reconstructions (*Figure 1—figure supplement 4*). Further, the sites that differed were a subset of those that differed between the ML and altAll reconstructions. Thus, the altAll reconstructions account for sequence changes due to both uncertainty given the ML tree and uncertainty due to topological uncertainty.

## Cloning and mutagenesis

All S100 genes in this study were purchased as synthetic constructs in pUC57 vectors from Genscript. S100 genes (A8s, A9s, A12s, MRP126s, and ancestrally reconstructed genes) were sub-cloned into a pETDuet-1 (pD) vector (Millipore). A8s, A12s, MRP126s, and ancCGs were cloned into multiple cloning site #1 (MCS1) of the pD vector, while A9s were cloned into MCS2. For expression and purification of A8/A9 heterocomplexes (A8/A9s), pD plasmids containing an A8 gene in MCS1 and an A9 gene in MCS2 were used as previously described (*Futami et al., 2016*). Opossum A8 was sub-cloned into an MBP-LIC vector to yield a His-MBP-TEV-opA8 construct. For opossum A8/A9, the entire His-MBP-TEV-A8 construct was then sub-cloned into MCS1 of a pD vector containing a marsupial A9 in MCS2. Other S100s (A1, A5, A7, A11, A14, and P) were previously cloned into a pET28/30 vector to yield a TEV-cleavable N-terminal His tag (*Wheeler et al., 2016*). Cysteine-free versions of all S100 genes, as well as point mutants, were prepared using site-directed mutagenesis (Agilent).

## Protein expression and purification

Recombinant protein overexpression was conducted in *E. coli* BL21 (DE3) pLysS Rosetta cells. Cultures were innoculated in luria broth overnight at 37°C, shaking at 250 rpm, in the presence of ampicillin and chloramphenicol. The following day, 10 ml of saturated culture was diluted into 1.5 L of media with antibiotics, grown to $OD_{600}$ = 0.6–1, and then induced overnight at 16°C using 1 mM IPTG. Cells were pelleted at 3,000 rpm for 20 min and stored at −20°C for no more than three months.

Lysates were prepared by vortexing pellets (3–5 g) in tris buffer (25 mM tris, 100 mM NaCl, pH 7.4) and incubating for 20 min at RT with DNAse I and lysozyme (ThermoFisher Scientific). Lysates were sonicated and cell debris was pelleted by centrifugation at 15,000 rpm at 4°C for >20 min. All proteins were purified on an Äkta PrimePlus FPLC using various 5 ml HiTrap columns (HisTrap FF (Ni-affinity), Q HP (anion exchange), SP FF (cation exchange), and MBPTrap HP (MBP) - GE Health Science). A1, A5, A7, A11, A14, and S100P were purified using a a TEV-cleavable His tag strategy used by our lab previously (*Lee et al., 2018*; *Postel et al., 2010*; *Gudmundsson and Agerberth, 1999*). All other S100s, except for opossum A8 and opossum A8/A9, were purified in three steps using Ni-affinity chromatography in the presence of calcium followed by two rounds of anion exchange chromatography at different pHs. For Ni-affinity chromatography, proteins were eluted over a 50 ml gradient from 25 to 1000 mM imidazole in tris buffer. Peak elution fractions were pooled and placed in dialysis overnight at 4°C in 4 L of tris buffer (calcium-free) adjusted to pH 8. Anion exchange chromatography was then performed the following day over a 50 ml gradient from 100 to 1000 mM NaCl in pH eight tris buffer. Fractions containing majority S100 were pooled and analyzed for purity on an SDS-PAGE gel. If trace contaminants remained, an additional anion exchange step was performed at pH six using the same elution strategy as for the previous anion exchange step.

Opossum A8 and A8/A9 lysates were prepared as above and then flowed over a nickel column, eluting over a 50 ml gradient from 25 to 1000 mM imidazole in tris buffer. Peak elution was pooled

and the MBP tag was cleaved by incubation with ~1:5 TEV protease at 4°C overnight in 4 L of tris buffer. The MBP tag was then removed by flowing the sample over an MBPTrap column, step-eluting with 10 mM maltose. Additional MBP columns were run until all MBP was removed from the purified protein, assessed by SDS-PAGE. If necessary, an additional anion exchange step at pH eight was performed to complete purification. All purified proteins were dialyzed overnight at 4°C in tris buffer + 2 g/L Chelex-100 resin (Biorad), flash-frozen the following day in liquid nitrogen, and stored at −80°C.

## Biophysical and biochemical characterization

For all experiments, protein aliquots were thawed fresh from freezer stocks and were either dialyzed in the appropriate experimental buffer overnight at 4°C or exchanged 3X into experimental buffer using 3K microsep spin concentrator columns (Pall Corporation). All samples were filter-sterilized using 0.1 µm spin filters (EMD Millipore) prior to measuring concentration and using in experiments. Thawed aliquots were used for no more than one week before discarding. All concentrations were measured by Bradford assay and correspond to micromolar dimeric protein.

For in vitro proteolytic susceptibility experiments, proteins were dialyzed or exchanged into tris buffer + 1 mM $CaCl_2$. 12.5 µM S100 protein was treated with 5 µM monomeric Proteinase K from *Tritirachium album* (Sigma Aldrich), cathepsin G from human neutrophils (Athens Research), or neutrophil elastase from human neutrophils (Millipore Sigma) in thin-walled PCR tubes, which were held at a constant temperature of 25°C over the course of the experiment using a thermal cycler. Proteinase K activity was quenched at different time points by directly pipetting an aliquot of the reaction into an equal volume of 95% Laemmli SDS-PAGE loading buffer + 5% BME at 95°C in a separate thermal cycler. Time points were analyzed via SDS-PAGE, and gels were quantified by densitometry using in-house gel analysis software (https://github.com/harmslab/gelquant, v1.0; copy archived at https://github.com/elifesciences-publications/gelquant; *Harman, 2020*). An exponential decay function ($A_o e^{-kt}$) was fit to the data to extract the decay rate, floating $A_o$ and $k$. Standard deviations were calculated from fits by taking the square root of the diagonalized covariance matrix and by error propagation.

Oligomeric states were measured using a superose 12 10/300 GL size exclusion column (Amersham Biosciences) with in-line concentration detection using refractive index (RI) and particle mass measured using a multiangle laser light scattering (MALS) instrument (Dawn Heleos, Wyatt Technology). Samples were concentrated to 0.5–2 mg/ml in tris buffer + 0.5 mM $CaCl_2$, 0.1 µm sterile-filtered, and analyzed at a flow rate of 0.2 ml/min. Data were processed using manufacturer's software (Astra).

Circular dichroism (CD) and chemical denaturation experiments were performed using a Jasco J-815 CD spectrometer and spectroscopy-grade guanidine hydrochloride (gdn-HCl) or urea. Chemical denaturation was performed using 25 µM dimeric protein in tris buffer with $CaCl_2$, with tris substituted for spectroscopy-grade trizma. Reversible unfolding and refolding curves were constructed by making concentrated 100 µM protein stocks in either buffer or 6M gdm or 10M urea and then preparing protein dilutions in various concentrations of gdn-HCl or urea in buffer. Samples were left to equilibrate in denaturant between three hours and overnight to allow for equilibration and were then analyzed by CD. Unfolding/refolding equilibration was confirmed by comparing unfolded vs. refolded protein at the same concentration. CD signal was quantified at 222 nm in a 1 mm cuvette using a 1 nm bandwidth, standard sensitivity, and 2 s D.I.T. HT voltage was <600 V. We fit a two-state unfolding model:

$$\frac{b_f + m_f x + (b_u + m_u x)e^{-\frac{G-mx}{RT}}}{1 + e^{-\frac{G-mx}{RT}}}$$

to the data to extract thermodynamic parameters, where $b_f$, $m_f$, $b_u$, and $m_u$ are the folded and unfolded baseline y-intercepts and slopes, $G$ is the unfolding free energy, $m$ is the m-value, $R$ = 0.001987 J·K$^{-1}$mol$^{-1}$ and $T$ = 298.15 K. Standard deviations were calculated from fits by taking the square root of the diagonalized covariance matrix and by error propagation. Apparent unfolding kinetics studies were performed using the above conditions by spiking concentrated protein stock directly into 6M gdm and immediately monitoring CD signal at 222 nm.

## Cell lines

We purchased commercially distributed HEK293T cells from ATCC (CRL-11268). Because we are using this cell line as a host for heterologous transient transfections, the appropriate control for consistency between assays is the measurement of reporter output for a set of control plasmids and a panel of known treatments. Upon thawing each batch of cells, we run a positive control for ligand-induced response. We transfect the cells with plasmids encoding human CD14, human MD-2, human TLR4, renilla luciferase behind a constitutive promoter, and firefly luciferase behind an NF-KB promoter. We then characterize the raw luciferase output for five treatments: 1) mock, 2) LPS, 3) LPS + polymyxin B, 4) S100A9 + polymyxin B, and 5) S100A9 + 1.25x polymyxin B. This has a stereotypical pattern of responses in renilla luciferase (high for all) and firefly luciferase (low, high, low, high, high). To validate that this response is dependent on the transfected TLR4 complex as opposed to the cells themselves, we repeat the experiment but exclude the TLR4 plasmid. This should give identical renilla luciferase values but no firefly luciferase output in response to any treatment. To ensure that the cells maintain their properties between passages, we repeat the mock, LPS, and LPS + polymyxin B control on every single experimental plate. This assay has a built-in control for mycoplasma contamination: high firefly luciferase signal in the absence of added agonist. This indicates that there is another source of TLR4-induced NF-kappa B output in the cells—most plausibly, contamination. This mycoplasma sensing approach is used in the commercially available HEK-BLUE mycoplasma detection kit (Invivogen). We discard any cells that exhibit high background values or reach 30 passages.

## Functional assays

The antimicrobial activity of S100s was measured against *S. epidermidis* using a well-established assay (**Hadley et al., 2018**; **Nakashige et al., 2016**; **Brunjes Brophy et al., 2013**; **Brophy et al., 2012**; **Cunden et al., 2016**). The day before, a 5 ml starter culture of *S. epidermidis* in tryptic soy broth (TSB) was grown overnight. The next day, the culture was diluted ~1:100 in TSB and grown for approximately 2 hr to an OD600 of ~0.8. Immediately prior to experiment, the *S. epidermidis* culture was again diluted 1:100 at a ratio of 62:38 experimental buffer (25 mM tris, 100 mM NaCl, 3 mM $CaCl_2$, pH 7.4):TSB. S100 proteins were exchanged into experimental buffer. Each well of a sterile 96-well plate was prepared with 40 µl of *S. epidermidis* diluted in experimental buffer + TSB, S100 protein at the desired concentration in experimental buffer, and then filled to 200 µl, maintaining a ratio of 62:38 experimental buffer:TB. *S. epidermidis* growth was monitored on a plate reader, measuring OD600 every 15 min for 13 hr. Each measurement was collected in technical triplicate and background-subtracted using a blank containing experimental buffer and TSB alone. Protein samples were confirmed to lack bacterial contamination by measuring S100 protein growth in experimental buffer and TSB lacking *S. epidermidis.*

All plasmids, cell culture conditions, and transfections for measuring the activity of S100s against TLR4s were identical to those previously described (**Nakashige et al., 2015**; **Liu et al., 2012**; **Loes et al., 2018**; **Anderson et al., 2019**). Briefly, human embryonic kidney cells (HEK293T/17, ATCC CRL-11268) were maintained up to 30 passages in Dulbecco's Modified Eagle Media (DMEM) supplemented with 10% fetal bovine serum (FBS) at 37°C with 5% CO2. Lipopolysaccharide *E. coli* K-12 LPS (LPS - tlrl-eklps, Invivogen) aliquots were prepared at 5 mg/ml in endotoxin-free water and stored at −20°C. Working solutions were prepared at 10 ug/ml and stored at 4°C to avoid freeze-thaw cycles. S100 proteins were prepared by exchanging into endotoxin-free PBS and incubating with an endotoxin removal column (Thermo Fisher Scientific) for 2 hr. S100 LPS contamination was assessed by measuring activity with and without Polymyxin B, an LPS chelating agent (*Figure 2—figure supplement 1*). LPS (200 ng per 100 µl well) or S100 (0.8, 0.4, 2, 4, or 5 µM dimer) treatments were prepared by diluting in 25:75 endotoxin-free PBS:serum-free Dulbecco's Modified Eagle Media (DMEM – Thermo Fisher Scientific). Polymyxin B (PB, 200 µg per 100 µl well) was added to all S100 experimental samples to limit background endotoxin contamination activity from recombinant protein preps. Cells were incubated with treatments for 3 hr prior to assaying activity. The Dual-Glo Luciferase Assay System (Promega) was used to assay Firefly and Renilla luciferase activity of individual wells. Each NF-κB induction value shown represents the Firefly luciferase activity divided by the Renilla luciferase activity, background-subtracted using the LPS + PB activity for each TLR4 species and normalized to the activity of LPS alone for each TLR4 species to normalize between plates. All

measurements were performed using three technical replicates per plate, a minimum of three biological replicates total, and a minimum of two separate protein preps.

For TLR4 activation measurements by A9 proteolytic products, 12.5 μM hA9 or hA9 M63F were incubated with 2.5 mg/ml Proteinase K immobilized to agarose at 37°C for increasing amounts of time. The reaction was quenched by spin-filtering the sample to remove Proteinase K. 2 μM A9 proteolysis treatments were then added to cells as outlined above. Western blots were performed by running an SDS-PAGE gel and transferring to a nitrocellulose membrane. Membranes were blocked using Odyssey Blocking Buffer for 1 hr, incubated with 1:1000 mouse anti-S100A9 primary antibody (M13 clone 1CD22, Abnova) for 1 hr, and incubated with 1:10,000 IRDye Goat anti-mouse 800CW IgG (H+L, Licor) for 1 hr, with 3 × 5 min TBST washes in between each step. Blots were imaged using the Licor Odyssey Fc imaging system.

## Species cartoons

All species cartoons were taken from the following websites: http://phylopic.org/image/c089caae-43ef-4e4e-bf26-973dd4cb65c5/, http://phylopic.org/image/aff847b0-ecbd-4d41-98ce-665921a6d96e/, http://phylopic.org/image/0f6af3d8-49d2-4d75-8edf-08598387afde/, http://phylopic.org/image/dde4f926-c04c-47ef-a337-927ceb36e7ef/. We acknowledge Sarah Werning and David Liao as authors of the opossum and mouse cartoons respectively, which were made publicly available through the creative commons attributions 3.0 unported license (https://creativecommons.org/licenses/by/3.0/).

## Acknowledgements

We thank current and former members of the Harms lab for helpful discussion and input. We thank the Barber lab for use of their cell culture facility. We thank Karen Guillemin for helpful comments on the manuscript.

## Additional information

### Funding

| Funder | Grant reference number | Author |
| --- | --- | --- |
| American Heart Association | 16 15BGIA22830013 | Michael J Harms |
| Pew Charitable Trusts | | Michael J Harms |
| National Institutes of Health | 3R01GM117140-03S1 | Michael J Harms |
| National Institutes of Health | T32GM007413 | Joseph L Harman Andrea N Loes |

The funders had no role in study design, data collection and interpretation, or the decision to submit the work for publication.

### Author contributions

Joseph L Harman, Conceptualization, Validation, Investigation, Visualization, Writing - original draft, Writing - review and editing; Andrea N Loes, Investigation, Visualization, Writing - review and editing; Gus D Warren, Maureen C Heaphy, Kirsten J Lampi, Investigation; Michael J Harms, Conceptualization, Data curation, Supervision, Funding acquisition, Visualization, Project administration, Writing - review and editing

### Author ORCIDs

Joseph L Harman (iD) https://orcid.org/0000-0002-8283-0301
Michael J Harms (iD) https://orcid.org/0000-0002-0241-4122

### Decision letter and Author response

Decision letter https://doi.org/10.7554/eLife.54100.sa1
Author response https://doi.org/10.7554/eLife.54100.sa2

## Additional files

### Supplementary files

• Supplementary file 1. Alignment of modern and ancestrally reconstructed S100 proteins used in this study.

• Supplementary file 2. Species-corrected tree used for ancestral sequence reconstruction.

• Transparent reporting form

### Data availability

All data generated or analysed during this study are included in the manuscript and supporting files.

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
