## [Decision Letter]

**Acceptance summary:**

This paper uses careful ancestral reconstruction work to provide a compelling case for how a single mutation in the S100A9 innate immunity protein produced its pro-inflammatory activity while also increasing proteolytic susceptibility, with the antimicrobial activity evolving independently. The results provide a clear example of how pleiotropy can be 'positive'. There are still relatively few studies in which such careful reconstruction work has been done so the work should be regarded as important and as a significant advance by a broad audience.

**Decision letter after peer review:**

Thank you for submitting your article "Evolution of multifunctionality through a pleiotropic substitution in the innate immune protein S100A9" for consideration by *eLife*. Your article has been reviewed by two peer reviewers, and the evaluation has been overseen by a Reviewing Editor and Wendy Garrett as the Senior Editor. The reviewers have opted to remain anonymous.

The reviewers have discussed the reviews with one another and the Reviewing Editor has drafted this decision to help you prepare a revised submission. *eLife*

Summary:

The reviewers each appreciated this well-written manuscript in which Harman et al. dissect the functional evolution of the S100A9 innate immune protein. Through careful ancestral reconstruction work, the authors provide a compelling case for how a single mutation produced the pro-inflammatory activity while also increasing proteolytic susceptibility, with the antimicrobial activity apparently independent. Together, the results provide a clear example of how pleiotropy can be 'positive'. There are still relatively few studies in which such careful reconstruction work has been done so the work was regarded as important and a significant advance.

Essential revisions:

1) The biggest claim for the study's impact relates to adaptive evolution under the potential constraint of multifunctionality. However, it wasn't clear to me how surprising the finding is given the location of the dimer interface relative to the M63F point mutation. Would there have to be a substantial allosteric effect to disrupt the interface? At least from the homodimeric A9 interaction illustrated in Figure 5A, it appears somewhat distant. Is that true of the antimicrobial heterodimer and/or its ability to coordinate and sequester metals? Related to this issue is the question of the functional impact of the pleiotropic M63F mutation. The notion of a proteolytic timer to control inflammatory signaling is intriguing, but is there any evidence that it has such a function or is it simply a coincidence with selection exclusive for TLR4 activation?

2) In general, the authors did a nice job of weighing evidence and balancing possible ambiguities in their results and different potential evolutionary scenarios that might break parsimony. However, it would be useful to have a better sense of how much uncertainty exists in the ancestral protein reconstructions (described in some places but not others). How many total substitutions are made compared to the modern variants and some help interpreting posterior probability values and the range of evolutionary space missing in the ancestral and alternate ancestral pairs.

3) The characterization of how additional point mutants influence S100A9 function could help to strengthen the argument that the proteolytic sensitivity and pro-inflammatory activities are coupled, but are relatively independent from the mutations influencing hA8/hA9 complex formation and antibacterial activity. However, this would add substantially more work, and it is unclear if it would dramatically reshape the main claims of the paper. Please comment. At a minimum, the authors should address in their responses and revision how much was already known, or not, about the functional coupling of the proteolytic and pro-inflammatory activities.

---

## [Author Response]

Essential revisions:1) The biggest claim for the study's impact relates to adaptive evolution under the potential constraint of multifunctionality. However, it wasn't clear to me how surprising the finding is given the location of the dimer interface relative to the M63F point mutation. Would there have to be a substantial allosteric effect to disrupt the interface? At least from the homodimeric A9 interaction illustrated in Figure 5A, it appears somewhat distant. Is that true of the antimicrobial heterodimer and/or its ability to coordinate and sequester metals? Related to this issue is the question of the functional impact of the pleiotropic M63F mutation. The notion of a proteolytic timer to control inflammatory signaling is intriguing, but is there any evidence that it has such a function or is it simply a coincidence with selection exclusive for TLR4 activation?

These are excellent points made by the reviewers that we have now expanded upon in the manuscript. We have addressed this question in two parts:

1a) Is it surprising that M63F doesn’t affect the A8/A9 complex despite its distance from the antimicrobial site and A8/A9 interface?

Given the location, the lack of effect is unsurprising. That said, the spatial separation of the sites – and the resulting lack of pleiotropy – is a result of our analysis, not a premise. We began this study wondering how multifunctionality could evolve given the possibility of pleiotropy.We did not know a priori which mutation(s) would be important for A9’s various functions, particularly since the interface through which A9 exerts proinflammatory activity is unknown. We identified the historical F63M substitution and showed that it affects both A9 proinflammatory activity and proteolytic resistance. Because position 63 is relatively far from the A8/A9 complex interface and metal sequestration site, it is unsurprising that mutating M63F has little effect on the A8/A9 complex. But we did not know where this mutation would be when we started the project.

We have altered the Results section where we describe this result to clarify this line of reasoning:

“A primary goal of this study was to understand the role of pleiotropy in the evolution of multifunctionality. […] These findings suggest that this single amino acid position had important effects on the evolution of A9 activation of TLR4 and loss of proteolytic resistance without significantly impacting A8/A9 oligomeric state, proteolytic resistance, or antimicrobial activity.”

1b) What is the functional significance, if any, of A9 losing proteolytic resistance? Is it simply coincident with A9 evolving proinflammatory activity?

To address this question, we have done a new set of experiments and made three modifications to the text:

To support the idea that A9 can be plausibly regulated by proteases found at sites of inflammation, we performed an experiment in which we tested both A9 and A9 M63F for proteolytic resistance against two common neutrophil proteases. These results are described in the Results:

“To relate these findings to proteases that A9 might encounter at sites of inflammation, we also measured the proteolytic resistance of human A9 and human A9 M63F against two neutrophil-specific proteases – cathepsin G and neutrophil elastase (Figure 4—figure supplement 2). […] We found that M63F decreased the rate of human A9 degradation in the presence of cathepsin G and neutrophil elastase in vitro by approximately one order of magnitude, matching our results using proteinase K (Figure 4—figure supplement 1).”

Second, we altered the Introduction to clarify what is known at this time:

“An additional layer of A9 immune function is that A9 and A8/A9 are thought to be regulated in the extracellular milieu by proteases. […] There is no obvious way to selectively increase the proteolytic resistance of A9 and test its effect on A9 activation of TLR4, making it difficult to understand the relationship, if any, between these two functions.”

Third, we now explicitly articulate three possible reasons for A9 loss of proteolytic resistance in the Discussion. The newly reorganized section reads:

“Why did A9s lose proteolytic resistance?

While proteolysis is not required for A9 activation of TLR4, it remains unclear why A9s lost proteolytic resistance. […] As A9s activate TLR4 in the protease-rich extracellular space, the functional result of A9 loss of proteolytic resistance is that A9s evolved a proteolytic “timer” concomitantly with evolving proinflammatory activity, all without affecting A8/A9 function.”

2) In general, the authors did a nice job of weighing evidence and balancing possible ambiguities in their results and different potential evolutionary scenarios that might break parsimony. However, it would be useful to have a better sense of how much uncertainty exists in the ancestral protein reconstructions (described in some places but not others). How many total substitutions are made compared to the modern variants and some help interpreting posterior probability values and the range of evolutionary space missing in the ancestral and alternate ancestral pairs.

We thank the reviewers for requesting these clarifications and have added the following to the Results section:

“AncA8 and ancA9 had average posterior probabilities of 0.88 and 0.83, with sequence similarities to human A8 and A9 of 66% and 64%, respectively (Figure 1—figure supplement 4). Average posterior probabilities in this range have been previously described as medium confidence reconstructions, with reconstructions characterized by others having average posterior probabilities as low as 0.7.”

We have also altered the text to explicitly state the number of sequence changes between the ML and AltAll reconstructions:

“AncA8/A9 and altancA8/A9 differ by 27 amino acids total (10 between ancA8 and altancA8 and 17 between ancA9 and altancA9 – Figure 1—figure supplement 4).”

And here:

“We used ASR to reconstruct the shared amniote ancestor of A9s, A8s, A12s, and MRP126s. […] We also constructed an alternate, “alt All” version of this ancestor (altancCG, S4), which differed from ancCG by 8 amino acids. The average posterior probability of ancCG was 0.86 (Figure 1—figure supplement 4).”

3) The characterization of how additional point mutants influence S100A9 function could help to strengthen the argument that the proteolytic sensitivity and pro-inflammatory activities are coupled, but are relatively independent from the mutations influencing hA8/hA9 complex formation and antibacterial activity. However, this would add substantially more work, and it is unclear if it would dramatically reshape the main claims of the paper. Please comment. At a minimum, the authors should address in their responses and revision how much was already known, or not, about the functional coupling of the proteolytic and pro-inflammatory activities.

These questions were a primary motivation for this work: to our knowledge, no previous study has characterized S100A9 mutations that alter both proteolytic resistance and proinflammatory activity. Previous work has shown A9 mutations that alter in vitro binding to TLR4 (Vogl et al., 2018); proteolytic fragments of A9 are sufficient to activate TLR4 (Vogl et al., 2018); proinflammatory stimuli stabilize A9 homodimers against proteolytic degradation (Riva et al., 2013); and mutating zinc-binding residues or removing the disordered C-terminal extension of A9 do not alter A9 activation of TLR4 (Loes et al. bioRxiv preprint). However, no studies to date have simultaneously examined mutations that affect both A9 proteolytic resistance and proinflammatory activity.

A challenge that we highlight in the Introduction is that there is no obvious way to selectively increase the proteolytic resistance of A9 to test its role in A9 proinflammatory activity. In this study we show that the M63F mutation both increases A9 proteolytic resistance and decreases proinflammatory activity, further suggesting a relationship between these two functions. We then showed that proteolysis is not required for A9 proinflammatory activity because 1) the A9 proteins do not get degraded in the proinflammatory assay, and 2) hA9 M63A, which is proteolytically susceptible, exhibits weaker proinflammatory activity than wildtype A9 (Figure 6).

To clarify previous work by others examining the relationship between proteolysis and A9 proinflammatory activity, we have altered the Introduction (see text changes in essential revision #1b). We have also altered the Discussion to clarify what we believe to be the functional relationship between A9 proteolysis and proinflammatory activity:

“While we cannot explicitly distinguish between each of these possibilities, the end result is that A9s lost proteolytic resistance from a resistant ancestor. As A9s activate TLR4 in the protease-rich extracellular space, the functional result A9 loss of proteolytic resistance is that A9s evolved a proteolytic “timer” concomitantly with evolving proinflammatory activity, all without affecting A8/A9 function.”